# Il-6 signaling exacerbates hallmarks of chronic tendon disease by stimulating reparative fibroblasts

**Tino Stauber[1], Greta Moschini[1,2], Amro A Hussien[1], Patrick Klaus Jaeger[1], Katrien De Bock[2], Jess G Snedeker[1]***

[1]Laboratory for Orthopedic Biomechanics, University Hospital Balgrist and ETH Zurich, Zurich, Switzerland; [2]Laboratory of Exercise and Health Department of Health Sciences and Technology (D-HEST) ETH Zurich, Swiss Federal Institute of Technology, Zurich, Switzerland

## eLife assessment

This **important** study defines signaling mechanisms in tendinopathy development, which is significant as there is a clear need to identify therapeutic targets to prevent or reverse tendon pathology. The evidence supporting the conclusions are **compelling** combining an existing human tendinopathy transcriptomics dataset with ex-vivo assembloid model, and an in vivo injury model using genetic reporter mice. This work will be of interest to developmental and stem cell biologists.

**\*For correspondence:**
jess.snedeker@hest.ethz.ch

**Competing interest:** The authors declare that no competing interests exist.

## Abstract

Tendinopathies are debilitating diseases currently increasing in prevalence and associated costs. There is a need to deepen our understanding of the underlying cell signaling pathways to unlock effective treatments. In this work, we screen cell signaling pathways in human tendinopathies and find positively enriched IL-6/JAK/STAT signaling alongside signatures of cell populations typically activated by IL-6 in other tissues. In human tendinopathic tendons, we also confirm the strong presence and co-localization of IL-6, IL-6R, and CD90, an established marker of reparative fibroblasts. To dissect the underlying causalities, we combine IL-6 knock-out mice with an explant-based assembloid model of tendon damage to successfully connect IL-6 signaling to reparative fibroblast activation and recruitment. Vice versa, we show that these reparative fibroblasts promote the development of tendinopathy hallmarks in the damaged explant upon IL-6 activation. We conclude that IL-6 activates tendon fibroblast populations which then initiate and deteriorate tendinopathy hallmarks.

## Introduction

Tendons are essential to every human movement (*Kirkendall and Garrett, 1997*). Tendinopathies represent the largest group of common tendon diseases and approximately 22% of all sport-related injuries (*Florit et al., 2019*). They can strike tendons at many different anatomical locations, can dramatically diminish quality of life by limiting the associated movements, and often share a history of repetitive overuse-induced damage and repair cycles (*Florit et al., 2019*; *McElvany et al., 2015*; *Snedeker et al., 2017*; *Yelin et al., 2016*). Once adult tendon regions fail to keep up with functional demands, they fall into a state of non-resolving, uncontrolled lesion repair (*Lipman et al., 2018*; *Riley, 2008*; *Soslowsky et al., 2000*; *Järvinen et al., 2005*; *Magnusson et al., 2010*; *Willett et al., 2007*; *Howell et al., 2017*).

These chronic tendon lesions underlying tendinopathies feature characteristics of normal wound healing including accelerated extracellular matrix (ECM) turnover and proliferation of reparative fibroblast populations as well as their migration to replace and repopulate damaged tissues (*Howell et al., 2017*; *Li et al., 2007*; *Millar et al., 2021*; *Gelberman et al., 1986*; *Sharma and Maffulli, 2006*; *Harvey et al., 2019*). In tendon, reparative (e.g. Scx⁺) fibroblast populations are assumed to reside primarily in the extrinsic compartment comprising epitenon and paratenon from where they are recruited to the damaged intrinsic compartment embodied by the load-bearing tendon core (*Gelberman et al., 1986*; *Dyment et al., 2013*; *Tan et al., 2021*; *Mendias et al., 2012*; *Mienaltowski et al., 2013*). While the mechanisms governing activation, proliferation, and recruitment of these populations are unclear, some insight can be gleaned from studies on other musculoskeletal tissues (*Snedeker et al., 2017*; *Gelberman et al., 1991*).

In acute muscle lesions, mechanisms for repair, hypertrophy, and hyperplasia are dominated by the satellite cells residing in the muscle basal lamina (*Serrano et al., 2008*; *Yin et al., 2013*; *Cosgrove et al., 2009*; *Ceafalan et al., 2014*). Satellite cells in muscle and dermal fibroblasts in skin are activated and recruited by interleukin-6 (IL-6), a key player in the acute phase response to stress (*Serrano et al., 2008*; *Ceafalan et al., 2014*; *Muñoz-Cánoves et al., 2013*). Stress-related mechanisms activating *IL-6* mRNA transcription in the absence of exogenous pathogens include damage-associated molecular patterns, calcium signaling after membrane depolarization, but also energetic stressors like glycogen depletion and redox signaling following exercise (*Tanaka et al., 2014*; *Kistner et al., 2022*). In humans, IL-6 transmits its signal via classical or trans-signaling (*Su et al., 2017*). Classical signaling involves the membrane-bound receptor IL-6R, which forms a homodimer with gp130 upon IL-6 binding. Trans-signaling works similarly, except that the IL-6R has been solubilized (sIL-6R) by metalloproteases (mostly ADAM10 and 17) (*Villar-Fincheira et al., 2021*), which cleave it from the cell membrane (*Su et al., 2017*; *Villar-Fincheira et al., 2021*). Since not all cell populations express IL-6R, trans-signaling via sIL-6R enables IL-6 signaling for a wider range of cell populations. Regardless of classical or trans-signaling initiation, further transduction of the IL-6 signal runs via two major pathways (JAK/STAT/ERK [*Zhang et al., 2013*; *Watanabe et al., 2004*] and SHP2/GAB2/MAPK [*Ernst and Jenkins, 2004*; *Eulenfeld et al., 2012*]) to turn on cellular processes inducing proliferation, migration, metabolic adaptations, and tissue turnover (*Su et al., 2017*; *Choy and Rose-John, 2017*). In chronic muscle lesions like Duchenne muscular dystrophy, IL-6 is persistently upregulated, and anti-IL-6 receptor antibodies have been proposed as treatment options (*Wada et al., 2017*). Anti-IL-6 receptor antibodies (IL-6 inhibitors) like tocilizumab inhibit both classical and trans-signaling and are routinely used in other chronic inflammatory diseases like systemic sclerosis, psoriasis, and rheumatoid arthritis (*Tanaka et al., 2014*; *Choy et al., 2020*; *Srirangan and Choy, 2010*; *Lewinson et al., 2018*; *Simone et al., 2003*). In this context, IL-6 inhibitors have been shown to reduce disease hallmarks including arthritis-concomitant tendon inflammation (*Poutoglidou et al., 2021*; *Choy et al., 2002*; *Emery et al., 2008*). Analogous to these chronic inflammatory musculoskeletal diseases, tendinopathies present with localized pain, swelling, and functional decline in the affected organ (*Snedeker et al., 2017*; *Riley, 2008*; *Colquhoun et al., 2022*; *Aström and Rausing, 1995*). Histological and molecular characteristics of tendinopathy include hypercellularity, disorganized collagen fibers including mechanically inferior collagen-3, and dysregulated ECM homeostasis (*Millar et al., 2021*; *Andersson et al., 2011*; *Jones et al., 2006*; *Riley et al., 1994*). Based on this research performed in other tissues, repair-competent fibroblasts appear as prime targets and effectors for IL-6 signaling in a tendon wound healing context (*Muñoz-Cánoves et al., 2013*; *Moresi et al., 2019*; *Nowell et al., 2003*; *McFarland-Mancini et al., 2010*).

Here, we investigated the hypothesis that IL-6 plays a vital role in activating reparative fibroblasts within the extrinsic tissue compartment (i.e. epitenon or paratenon) and recruitment of these cells to the damaged tendon core tissue in non-sheathed tendons (*Howell et al., 2017*; *Stauber et al., 2021*; *Sakabe et al., 2018*). While this likely represents a critical step in normal tendon healing (*Lin et al., 2006*; *Nakama et al., 2006*; *Stauber et al., 2020*), we propose that extended and excessive IL-6 signaling may causally exacerbate tendinopathy in non-sheathed tendons (*Legerlotz et al., 2012*).

We experimentally tested and confirmed these hypotheses in four steps:

1. We showed that the IL-6/JAK/STAT signaling cascade is positively enriched in (non-sheathed) human tendinopathic tendons alongside gene signatures typical for fibroblasts as well as

downstream gene sets suggesting excessive cell proliferation (hypercellularity), imbalanced ECM turnover, and neo-vascularization.

2. We verified increased IL-6 and IL-6R levels in tendinopathic human tissue samples with fluorescence microscopy. We also co-stained these sections with a marker for reparative fibroblasts (CD90[+]) to confirm their increased presence and gauge their contribution to the elevated IL-6 and IL-6R levels in tendinopathic human tissues.

3. We exploited an explant-based assembloid model system to confirm the causal effect of IL-6 signaling on extrinsic fibroblast activation, recruitment, and proliferation in tendinopathic niche conditions.

4. We followed the downstream effects of enhanced extrinsic fibroblast activation and accumulation on the tendon core embedded in our assembloids. Here, we document the emergence of central tendinopathic hallmarks including aberrant (catabolic) matrix turnover, hypercellularity, and hypoxic responses with a stronger fibroblast presence, which is reduced when IL-6 core signaling is inhibited.

## Results

### The IL-6/JAK/STAT signaling signature is positively enriched in human tendinopathic tendons alongside signatures of extrinsic cell population activation and hallmarks of clinical tendinopathy

To better illuminate the ongoing wound healing processes that are a central feature of chronic tendon disease, we first searched for enriched signaling pathways underlying tendinopathic tendons in a publicly available dataset (GEO: GSE26051) (*Jelinsky et al., 2011*). To deduce common disease patterns affecting tendons from diverse anatomical locations, the microarray dataset contained analyzed samples from 23 normal and 23 tendinopathic human tendons of mixed anatomical origin. We excluded 5 normal and 5 tendinopathic samples from sheathed tendons originally included in the dataset, as those tendons are organized into different sub-compartments than the non-sheathed tendons remaining in the dataset (*Figure 1—figure supplement 1*, *Supplementary file 1*). Overall, we found 240 significantly upregulated genes, 474 significantly downregulated genes, and 20,359 not-differentially regulated genes in tendinopathic compared to normal control tendons.

Further analysis revealed significantly upregulated *IL6* and *IL6* signaling transducer (*IL6ST,* also known as *GP130*) transcripts in tendinopathic tendon tissue (*Figure 1A and B*, *Table 1*). Conversely, *IL6* receptor (*IL6RA*) expression trended toward downregulation. Focusing on proteases that generate soluble *IL6RA*, we found significant upregulation of both *ADAM10* and *ADAM17* in human tendinopathic tendons. Downstream of *IL6* receptor binding, *JAK1* trended toward upregulation and *STAT3* was upregulated significantly. Other *IL6* regulated signaling checkpoints such as *MAPK3* and *GAB2* trended toward downregulation. Aside from *IL6*, another significantly upregulated member of the IL-6 family was interleukin-11 (*IL11*).

In line with aberrant matrix turnover generally featured in tendinopathy, the transcripts of the following genes were significantly increased in the tendinopathic samples: *COL1A1*, *COL1A2*, *COL18A1*.

Since changes in single transcripts alone have a limited predictive value for pathway-level changes, we next performed unbiased gene set enrichment analysis (GSEA) using the human hallmark dataset from MSigDB (*Subramanian et al., 2005*; *Liberzon et al., 2015*). Confirming the trends from the single transcript analysis, GSEA revealed a positive enrichment of the IL-6/JAK/STAT pathway (q-value: 0.003) alongside gene sets matching well-known tendinopathy hallmarks such as neo-vascularization (i.e. angiogenesis, mTORC1 signaling) and hypercellularity (i.e. G2M checkpoint, mitotic spindle, MYC targets v1, epithelial mesenchymal transition, and E2F targets) in the tendinopathic samples compared to the normal controls (*Figure 1C and D*). We then looked further into aberrant biological processes by mapping the significantly changed single transcripts (p-value<0.01) to the respective gene ontology (GO) database in an overrepresentation analysis (ORA) (*Figure 1E*, *Figure 1—figure supplement 2*, *Figure 1—figure supplement 3*). The emerging processes pointed toward ongoing morphogenesis and wound healing favoring hypercellularity (i.e. proliferation, migration) and ECM turnover, which are both established hallmarks of tendinopathy. Lastly, we matched the detected transcript changes to the human cell-type signature gene sets from MSigDB in a GSEA to estimate the contribution of fibroblasts to the aberrant processes (*Subramanian et al., 2005*; *Liberzon et al.,*

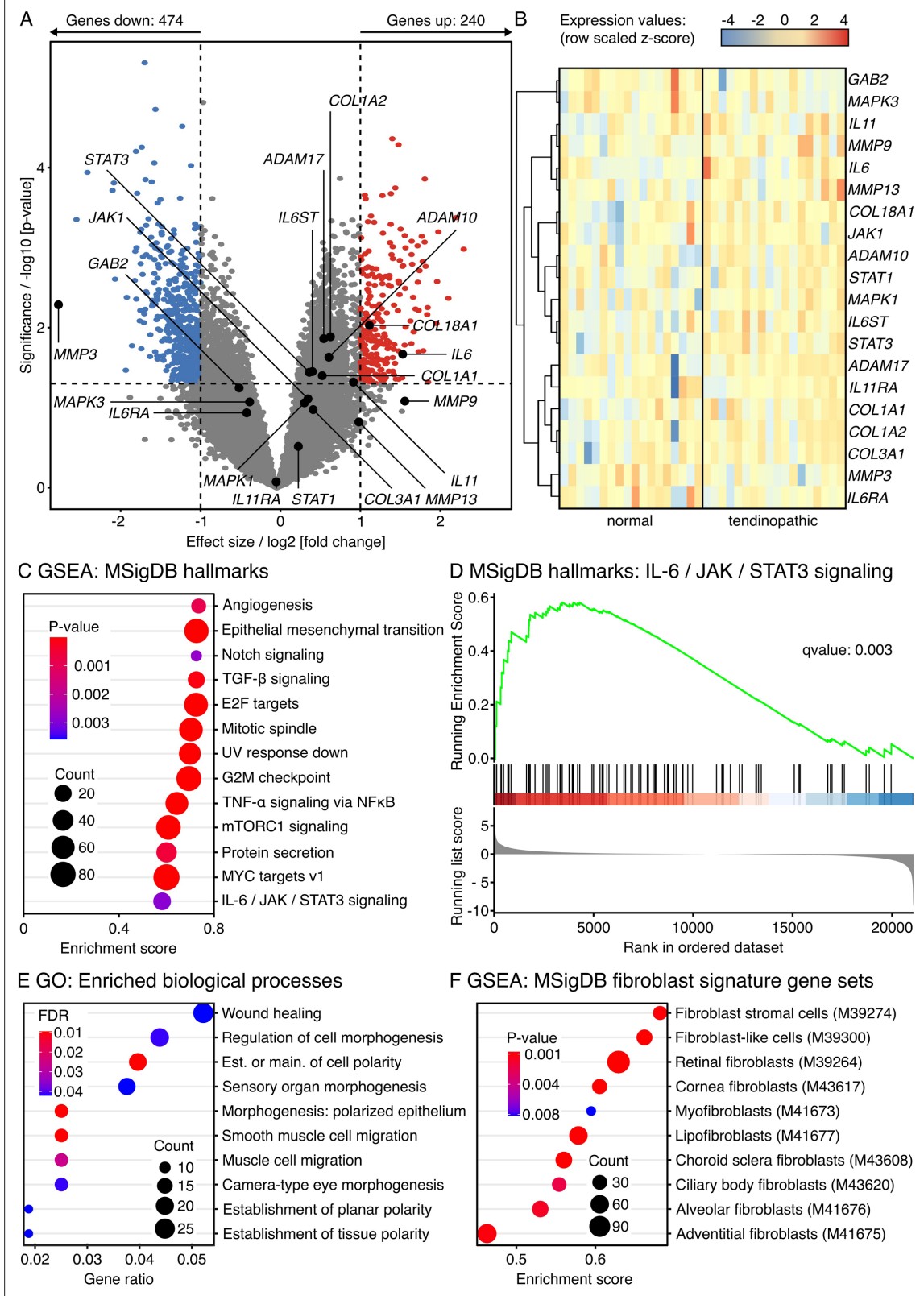

**Figure 1.** Transcriptome analysis of up- and downregulated genes and pathways in tendinopathic compared to normal control human tendons (non-sheathed). (**A**) Volcano plot of differentially expressed genes (DEGs) comparing tendinopathic to normal human tendons. Genes colored in red have a log2 (fold change)>1, a p-value<0.05, and are therefore considered to be significantly increased in tendinopathic tendons. Genes colored in blue have a log2 (fold change)<−1, a p-value<0.05, and are therefore considered to be significantly decreased in tendinopathic tendons. The log2 and p-value

*Figure 1 continued on next page*

*Figure 1 continued*

thresholds are represented by the dashed lines. Annotated genes are part of the IL-6 cytokine superfamily, the IL-6 signaling cascade, or involved in matrix turnover. (**B**) Unsupervised hierarchical clustering of expression values from members of the IL-6 cytokine superfamily, their receptors, and parts of the IL-6 signaling cascade (biological replicates: N=18 normal, N=18 tendinopathic). Genes are clustered by color with positive (red) or negative (blue) row-scaled z-scores. (**C**) Dotplot showing significantly enriched gene sets (p-value<0.05) as determined by gene set enrichment analysis (GSEA) based on the MSigDB human hallmark gene sets. The color of the circles represents their p-value, the size the number of enriched genes (count), and the position on the x-axis the enrichment score as well as its direction. (**D**) GSEA plot for the IL-6/JAK/STAT3 signaling hallmark contained in the MSigDB human hallmark gene sets. The green line traces the running enrichment score on the y-axis while going down the rank of genes listed on the x-axis, the black lines standing in blue and red bars indicate the locations of the genes related to the pathway in the ranked list, and the gray histogram shows the running list score across the ranks. (**E**) Dotplot showing the top 10 gene ontology (GO) gene sets for biological processes most significantly enriched by overlapping DEG sets. The color of the circles represents their adjusted p-value (false discovery rate [FDR]), the size the number of enriched genes (count), and the position on the x-axis the number of enriched genes in ratio to the total number of genes annotated to the gene set (gene ratio). (**F**) Dotplot showing significantly enriched fibroblast signature gene sets (p-value<0.05) as determined by GSEA based on the MSigDB human cell-type signature gene sets. The color of the circles represents their p-value, the size the number of enriched genes (count), and the position on the x-axis the enrichment score as well as its direction.

The online version of this article includes the following figure supplement(s) for figure 1:

**Figure supplement 1.** Principal component analysis (PCA) plots of the human tendon microarray data.

**Figure supplement 2.** Detailed transcriptome analysis of up- and downregulated genes and pathways in normal and human tendinopathic tendons (non-sheathed).

**Figure supplement 3.** Detailed transcriptome analysis of up- and downregulated genes and pathways in normal and human tendinopathic tendons (non-sheathed).

**Table 1.** Effect sizes and p-values for selected transcripts.
The data describes the differences between tendinopathic and normal control human tendons (non-sheathed).

| Transcript | Effect size | p-Value |
|---|---|---|
| IL6 | 1.529 | 0.021 |
| IL6RA | –0.423 | 0.116 |
| IL6ST / GP130 | 0.399 | 0.035 |
| ADAM10 | 0.606 | 0.023 |
| ADAM17 | 0.541 | 0.014 |
| JAK1 | 0.344 | 0.078 |
| STAT1 | 0.224 | 0.305 |
| STAT3 | 0.36 | 0.036 |
| MAPK1 | 0.3 | 0.086 |
| MAPK3 | –0.386 | 0.085 |
| GAB2 | –0.517 | 0.057 |
| IL11 | 0.913 | 0.048 |
| IL11RA | –0.053 | 0.844 |
| COL1A1 | 0.523 | 0.04 |
| COL1A2 | 0.624 | 0.013 |
| COL3A1 | 0.408 | 0.106 |
| COL18A1 | 1.112 | 0.009 |
| MMP9 | 1.556 | 0.083 |
| MMP13 | 0.981 | 0.151 |
| MMP3 | –2.779 | 0.005 |

*2015*). While this database does not yet include tendon-specific fibroblast populations, the signature gene sets of several fibroblast populations were significantly enriched by the transcript changes detected in tendinopathic tendons (*Figure 1F*). To confirm the increase of transcripts related to IL-6 signaling and fibroblast presence on the protein level, we next assessed human patient samples using fluorescence microscopy.

## IL-6, IL-6R, and CD90 are elevated on the protein level in tendinopathic human tendons compared to normal control tendons

In a second step, we sought to validate the gene array analysis highlighting elevated IL-6-IL-6R signaling as well as the elevated presence of fibroblasts in non-sheathed tendinopathic tendons compared to non-sheathed normal control tendons on the protein level. We thus extracted tissue sections from tendinopathic biceps tendons and from normal control tendons leftover after anterior cruciate ligament reconstruction surgery (*Figure 2A* and *Figure 2—figure supplement 1*) and stained them with fluorescently labeled IL-6, IL-6R, and CD90 antibodies.

In normal control tendons, the fluorescent signal stemming from the IL-6 antibody (*Figure 2B*, left side, red) appeared to be confined to the extrinsic compartment, which we identified based on the clustering of cells with a roundish nucleus (blue). In tendinopathic tendons (*Figure 2B*, right side), it was challenging to identify the extrinsic compartment due to the characteristic change in cell shape from elongated to more roundish in both the extrinsic compartment and the load-bearing core tissue. While the signal of the IL-6 antibody was more evenly distributed over the tendinopathic tissue section and tendon compartments compared to the control, it was still more prominent around roundish than elongated cells. Using the nuclear staining as a mask, we attributed IL-6 secretion to cells based on spatial proximity. The percentage of IL-6 secreting cells was only slightly increased in tendinopathic compared to healthy control tendons (*Table 2*).

The cell surface protein CD90 is a common marker of reparative fibroblasts (*Ho et al., 2019*; *Li et al., 2021*). Here, we visually detected its signal on only a few cells in the normal control tendons (*Figure 2C*, left side, green) but on a large number of cells in the tendinopathic tendons (*Figure 2C*, right side). The subsequent spatial proximity-based quantification confirmed this initial visual impression by detecting a statistically significant difference in the percentage CD90$^+$ in the normal control compared to the tendinopathic tendon (*Table 2*).

The IL-6 receptor (IL-6R) is another central part of the IL-6 signaling cascade. While some cells in the normal control tendons stained positively for IL-6R, many more seemed to be present in the tendinopathic tendons. We could confirm this again with spatial proximity-based quantification detecting a statistically significant difference in the percentage of IL-6R$^+$ cells in normal control compared to tendinopathic tendons (*Table 2*).

## Both IL-6 and IL-6R appear in spatial proximity to CD90$^+$ and CD68$^+$ cells in non-sheathed human tendinopathic tendons

In another study conducted in mouse tendons, immune cells such as macrophages were reported as major sources of IL-6R during tendon growth (*Bautista et al., 2023*). We therefore co-stained cells with IL-6, IL-6R, and the established human macrophage surface marker CD68 to see whether this was also true in the context of human tendinopathic tendons. To further check whether (reparative) fibroblasts could indeed be involved in IL-6-IL-6R signaling as initially hypothesized, we also co-stained human tendinopathic tendons with CD90, IL-6, and IL-6R. We again used tendinopathic tissues extracted from diseased biceps tendons (*Figure 3A* and *Figure 2—figure supplement 1*).

We found a high percentage of cells in close spatial proximity to fluorescent signals generated by the IL-6 antibodies (*Figure 3B*, left side, red) to be CD90$^+$ (green and blue), identifying them as a likely source of IL-6. Again, IL-6 secreting, CD90$^+$ cells assumed a more roundish phenotype in contrast to the CD90$^+$ cells not secreting IL-6. The overlap between signals from CD68$^+$ cells (*Figure 3B*, right side, green and blue) and the IL-6 antibodies (red) seemed less pronounced. Indeed, quantification of the spatial signal overlay showed a significantly higher percentage of cells in spatial proximity to IL-6 to be CD90$^+$ rather than CD68$^+$ cells (*Table 3*).

The presence of IL-6R on different cell populations could provide cues on the targets of IL-6 signaling in tendinopathic tendons. In the tendinopathic sections probed here, almost all cells that stained positively for IL-6R (*Figure 3C*, left side, magenta) were also CD90$^+$ (green and blue) but less

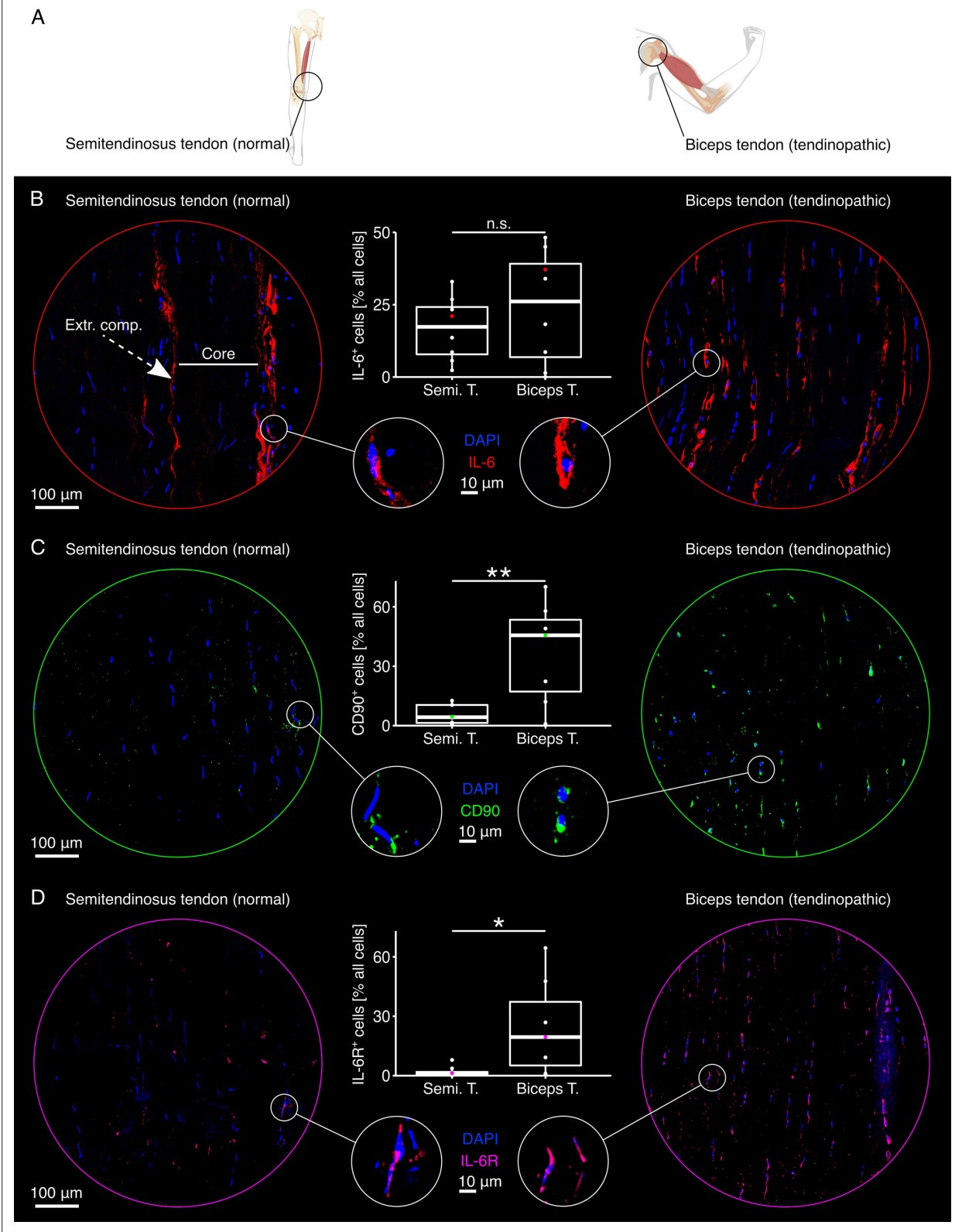

**Figure 2.** Distribution of IL-6, CD90, and IL-6R in normal control and tendinopathic human tendons (non-sheathed). (**A**) Illustrative depiction of the origins of the tendons used in this experiment. Normal control tendon tissues were taken from semitendinosus and gracilis tendons leftover from anterior cruciate ligament (ACL) reconstruction surgery. Tendinopathic tissues were taken from painful shoulders during surgery (**Figure 2—figure supplement 1**). (**B**) Representative fluorescence microscopy images of normal control (left) and tendinopathic tendons (right) stained with DAPI (blue)

*Figure 2 continued on next page*

*Figure 2 continued*

and an IL-6 antibody (red). Boxplots depict the quantified co-localization of DAPI and IL-6 (IL-6+ cells) calculated as percentage of the total number of cells. Biological replicates: N=8. (**C**) Representative fluorescence microscopy images of normal control (left) and tendinopathic tendons (right) stained with DAPI (blue) and a CD90 antibody (green). Boxplots depict the quantified co-localization of DAPI and CD90 (CD90+ cells) calculated as percentage of the total number of cells. Biological replicates: N=7. (**D**) Representative fluorescence microscopy images of normal control (left) and tendinopathic tendons (right) stained with DAPI (blue) and an IL-6R antibody (magenta). Boxplots depict the quantified co-localization of DAPI and IL-6R (IL-6R+ cells) calculated as percentage of the total number of cells. Biological replicates: N=7. In all boxplots, each datapoint was calculated from eight representative fluorescence microscopy images taken from the same sample. The colored datapoint matches the presented fluorescence microscopy image. The upper and lower hinges correspond to the first and third quartile (25th and 75th percentile), the middle one to the median, the whiskers extend from the hinges no further than 1.5 times the interquartile range, and the points beyond the whiskers are treated as outliers. Results of the statistical analysis are indicated as follows: $^{n.s.}p \geq 0.05$, $^{*}p<0.05$, $^{**}p<0.01$. The applied statistical test was the Student's t-test.

The online version of this article includes the following figure supplement(s) for figure 2:

**Figure supplement 1.** Metadata of the human patient-derived tissues analyzed with fluorescence microscopy.

than half of them were CD68+ cells (*Figure 3C*, right side, green and blue). Quantifying the difference based on spatial proximity confirmed this impression (*Table 3*) and the statistical analysis judged it to be statistically significant. The staining antibody and quantification method deployed here likely cannot discriminate between IL-6R produced by the cell carrying it and IL-6R that was solubilized participates in trans-signaling.

We conclude from the above analysis that IL-6 signaling in non-sheathed tendinopathic tendons plausibly contributes to chronic tendinopathic hallmarks like hypercellularity and aberrant matrix turnover, potentially by activating reparative (CD90+) fibroblast populations through IL-6R. On this basis, we sought to directly test whether a causal relationship exists between the observed changes in IL-6 signaling and the associated disease processes. To this end, we harnessed an in vitro assembloid model of inter-compartmental crosstalk to better dissect the role of IL-6 in tendinopathy.

## IL-6 signaling by tendon core explants activates extrinsic fibroblasts

We have previously validated a hybrid explant // hydrogel assembloid model that reproduces the in vivo tissue compartment interface between the load-bearing tendon core and the extrinsic compartment (i.e. epitenon and paratenon) of non-sheathed tendons (*Stauber et al., 2021*; *Stauber et al., 2024*). We exploited this model to test whether IL-6 signaling across tissue compartments could activate fibroblast populations in the peritendinous space in a manner that mimics the IL-6 signaling signatures we uncovered in the human data analysis (*Figure 4A*). Briefly, we isolated and clamped mouse tail tendon fascicles to represent the tendon core while selecting (mainly Scx+ and CD146+) fibroblasts from digested Achilles tendons based on plastic adherence growth and surface marker expression as established previously and repeated here (*Figure 4B*, *Figure 4—figure supplement 1*, *Figure 4—figure supplement 2*; *Stauber et al., 2021*; *Tarafder et al., 2017*). To form the artificial extrinsic compartment, we encapsulated these fibroblast populations into a collagen hydrogel which we then let polymerize around the clamped core explants (*Figure 4C*).

In a separate experiment, we verified the presence and located the cellular sources of IL-6 and IL-6R using supernatant and flow cytometric analysis respectively (*Figure 4—figure supplement 3*). Both were present in the supernatant (IL-6) and on CD45+ cell populations (IL-6R) from core explants, but not in the supernatant or on the surface of extrinsic fibroblasts cultured in a collagen hydrogel. This indicates that the following described effects of IL-6 on extrinsic fibroblasts could be dominated by trans-signaling.

To dissect the effect of IL-6 signaling on the extrinsic target populations, we integrated either wildtype-derived (WT) or IL-6 knock-out-derived (KO) explants from the B6.129S2-Il6tm1Kopf/J (*Kopf*

**Table 2.** Percentages of IL-6+, CD90+, and IL-6R+ cells of all cells in tendinopathic and normal control tissues derived from human patients.
The values are given as median(IQR).

| Condition | IL-6+ % all cells | CD90+ % all cells | IL-6R+ % all cells |
|---|---|---|---|
| Tendinopathic | 33.3 (38.4) | 45.6 (36.3) | 37.5 (41.2) |
| Normal control | 22.5 (17.2) | 4.3 (9.0) | 5.3 (10.2) |

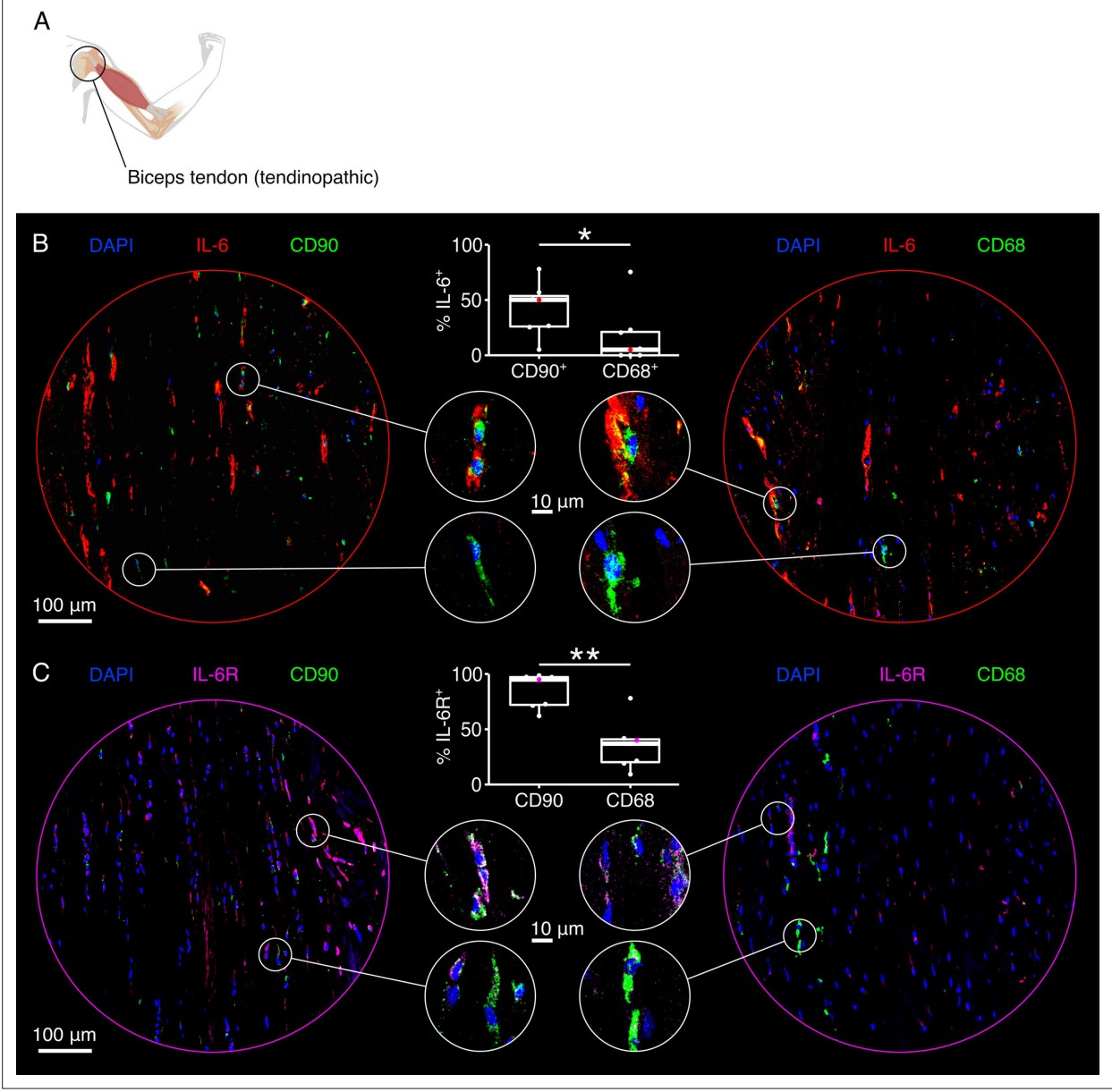

**Figure 3.** Co-localization of IL-6, IL-6R, CD90, and CD68 in tendinopathic human tendons (non-sheathed). (**A**) Illustrative depiction of the origin of the tendon used in this experiment (painful biceps tendons, *Figure 2—figure supplement 1*). (**B**) Representative fluorescence microscopy images of tendinopathic tendons stained with DAPI (blue), an IL-6 antibody (red), and either a CD90 antibody (left image, green) or a CD68 antibody (right image, green). Boxplots depict the quantified co-localization of DAPI, IL-6, and either CD90 (biological replicates: N=7) or CD68 (biological replicates: N=8) calculated as percentage of the number of DAPI⁺IL-6⁺ cells. (**C**) Representative fluorescence microscopy images of tendinopathic tendons stained with DAPI (blue), an IL-6R antibody (magenta), and either a CD90 antibody (left image, green) or a CD68 antibody (right image, green). Boxplots depict the quantified co-localization of DAPI, IL-6, and either CD90 or CD68 calculated as percentage of the number of DAPI⁺IL-6R⁺ cells (biological replicates: N=7). In all boxplots, each datapoint was calculated from eight representative fluorescence microscopy images taken from the same sample. The colored datapoint matches the presented fluorescence microscopy image. The upper and lower hinges correspond to the first and third quartile (25th and 75th percentile), the middle one to the median, the whiskers extend from the hinges no further than 1.5 times the interquartile range, and the points beyond the whiskers are treated as outliers. Results of the statistical analysis are indicated as follows: *p<0.05, **p<0.01. The applied statistical test was the Mann-Whitney-Wilcoxon test.

**Table 3.** CD90+ and CD68+ cells as percentages of IL-6+ and IL-6R+ cells in tendinopathic tissues derived from human patients. The values are given as median(IQR).

|  | CD90+ (median(IQR)) | CD68+ (median(IQR)) |
|---|---|---|
| % of IL-6+ | 50.2 (27.6) | 5.02 (21.1) |
| % of IL-6R+ | 95.0 (25.1) | 37.0 (20.6) |

et al., 1994) mouse line (*Figure 5A*). We then performed bulk RNA-sequencing (RNA-seq) on the extrinsic populations after 1 week of co-culture in tendinopathic niche conditions (*Stauber et al., 2021*; *Blache et al., 2021*; *Wunderli et al., 2020*).

Overall, integration of an IL-6 KO core increased transcripts of 256 genes in the surrounding extrinsic compartment, decreased transcripts of 98 genes, and left 15,295 unchanged (*Figure 5B and C*). After mapping the significant transcript changes to biological processes in the GO database, we conducted a biased search for processes matching those dysregulated in human tendinopathic

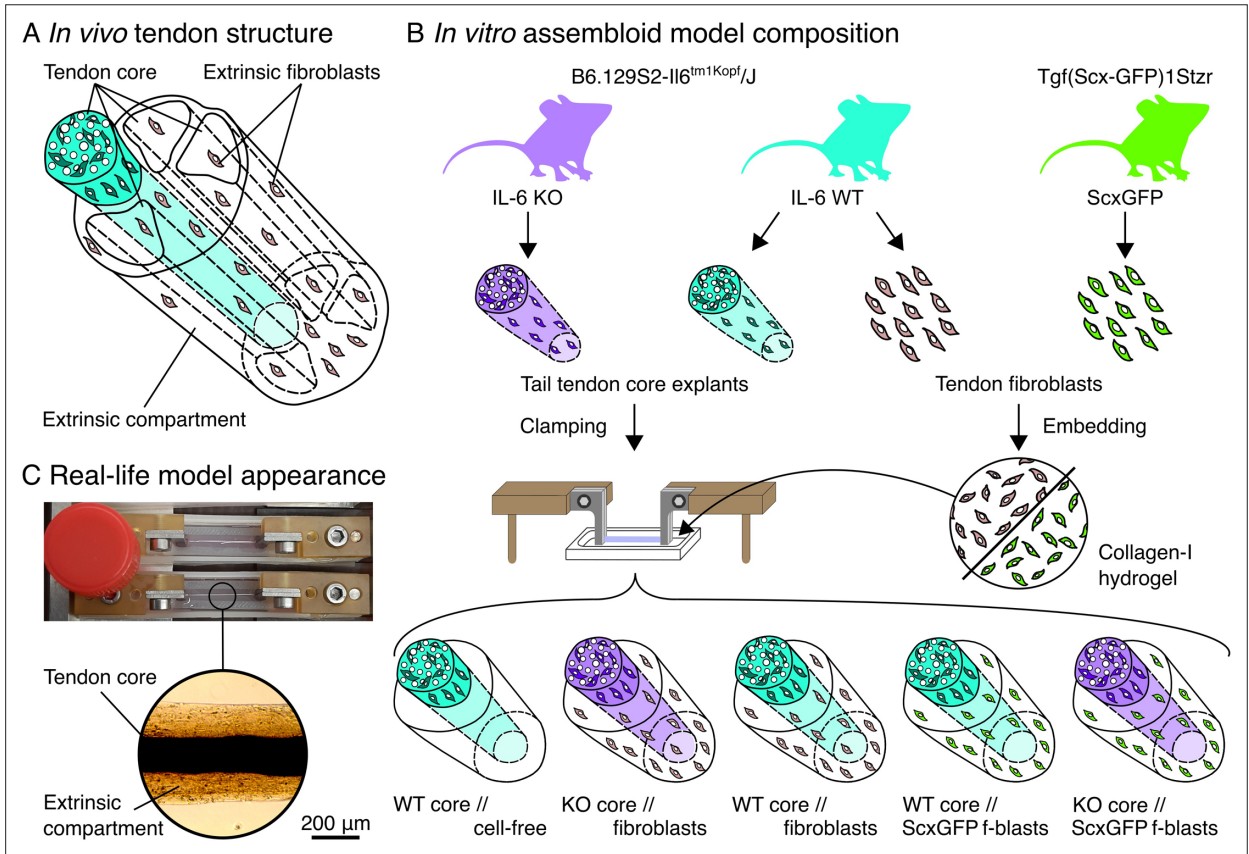

**Figure 4.** Concept behind the in vitro hybrid explant // hydrogel assembloid system. (**A**) Abstract representation of the in vivo load-bearing tendon core subunits (light blue/white) surrounded by the extrinsic compartment (white) containing, i.e., extrinsic fibroblasts (light brown). (**B**) Sources of the in vitro model system components with the IL-6 knock-out core (KO core) in violet, the IL-6 wildtype core (WT core) in light blue, the IL-6 wildtype fibroblasts in light brown, and the ScxGFP fibroblasts in green. Core explants were clamped, and the fibroblasts embedded in a (liquid) collagen solution before crosslinking the mixture into a hydrogel around the clamped core explants in various combinations. (**C**) Photographic and light microscopic images of the in vitro assembloid model system. Lid of a 15 ml Falcon tube (∅: 17 mm) used for scale.

The online version of this article includes the following figure supplement(s) for figure 4:

**Figure supplement 1.** Characterization of cell populations derived from Achilles tendons and tail tendon fascicles of ScxGFP mice (morphology and surface markers).

**Figure supplement 2.** Characterization of cell populations derived from Achilles tendons and tail tendon fascicles of ScxGFP mice (gene transcripts).

**Figure supplement 3.** Identification of cellular IL-6 and IL-6R sources in mouse tendon assembloids using flow cytometry and analyzing the supernatant.

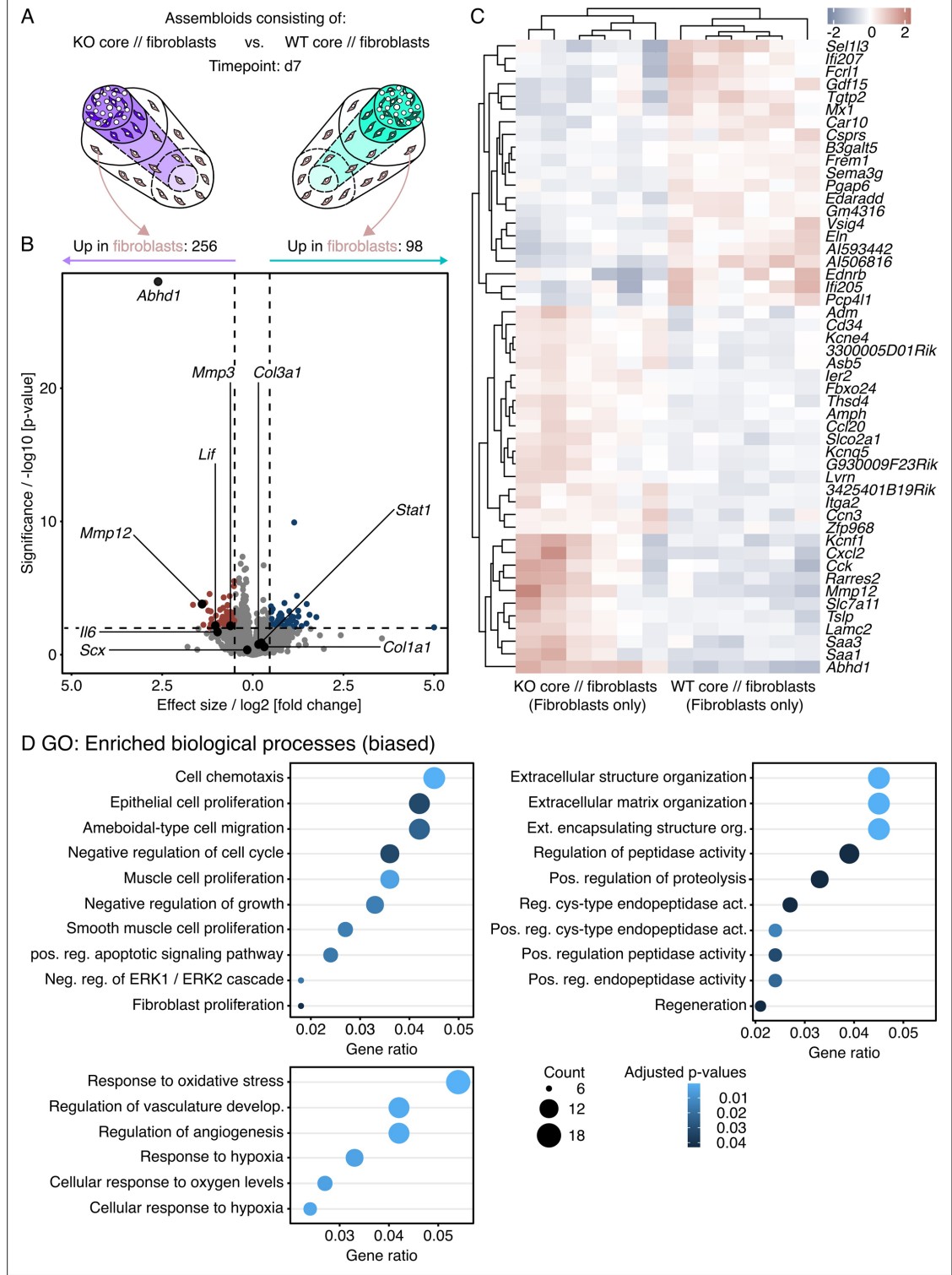

**Figure 5.** Transcript changes in hydrogel-embedded fibroblasts seeded around an IL-6 knock-out (KO) core explant compared to those seeded around a wildtype (WT) core. (**A**) Illustration of the assembloid combinations compared here (KO core // fibroblasts vs. WT core // fibroblasts), the assessed timepoint (**d7**), and the analyzed compartment (extrinsic fibroblasts only). (**B**) RNA-seq volcano plot of differentially expressed genes (DEGs). Genes colored in red have a log2 (fold change)>0.5, a p-value<0.05, and are considered to be significantly increased in the extrinsic compartment of KO core

*Figure 5 continued on next page*

*Figure 5 continued*

// fibroblast assembloids. Genes colored in blue have a log2 (fold change)<–0.5, a p-value<0.05, and are considered to be significantly increased in the extrinsic compartment of WT core // fibroblast assembloids. The log2 and p-value thresholds are represented by the dashed lines. (C) Unsupervised hierarchical clustering of the top 50 DEGs. Genes are clustered by color with positive (red) or negative (blue) row-scaled z-scores. Columns represent individual samples (biological replicates: N=6). (D) Dotplots depicting a selection of gene ontology (GO) annotations significantly enriched (adjusted p-value<0.05) by the DEGs. The selection was biased by GO biological process annotations enriched in the human dataset (*Figure 1E*). The color of the circles represents their adjusted p-value, the size the number of enriched genes (count), and the x-axis the number of enriched genes in ratio to the total number of genes annotated to the gene set (gene ratio).

The online version of this article includes the following figure supplement(s) for figure 5:

**Figure supplement 1.** Detailed transcriptome analysis of genes up- and downregulated in hydrogel-embedded fibroblast seeded around an IL-6 knock-out (KO) core explant compared to those seeded around a wildtype (WT) core.

tendons. Out of the 195 significantly enriched biological processes (adjusted p-value<0.05) in the extrinsic compartment of KO core // fibroblast assembloids, 10 (5.1%) could be linked to changes in local cellularity and another 10 to ECM turnover (*Figure 5D*).

When stratifying the respective contribution of transcripts increased and decreased around a KO core compared to a WT core to the top 20 enriched GO biological processes and significantly enriched MSigDB mouse hallmarks, we found that IL-6 signaling by the core correlated positively with processes aimed at increasing cellularity and ECM turnover (*Figure 5—figure supplement 1B–D*), which are both hallmarks of tendinopathic tissues and indicators of an activated tissue state.

To verify the transcript-level differences connected to hypercellularity also on the tissue level, we next performed a proliferation and migration analysis in our assembloid model system.

## IL-6 signaling by tendon core explants stimulates cell proliferation and Scx⁺ cell recruitment to the signaling tendon core

Using the assembloid model, we investigated whether IL-6 signaling could play a causative role in the hypercellularity that is a major hallmark of tendinopathy. Closely mimicking human tendinopathic tendons, we indeed found cellularity-increasing biological processes to be positively enriched in cell populations around WT compared to those around IL-6 KO tendon core explants. We then assessed whether these IL-6-dependent transcript-level changes would translate to an increased cell density. To do this, we seeded tendon fibroblast populations isolated from ScxGFP mice (co-expressing the tendon marker scleraxis [Scx] with a green fluorescent protein) into the hydrogel extrinsic compartment of our assembloids and incorporated either a WT or an IL-6 KO core into the center (*Figure 6A*, top panels).

Representative fluorescence microscopy images taken after 7 days in co-culture confirmed a higher total cell number in WT core // ScxGFP fibroblast assembloids compared to KO core // ScxGFP fibroblast assembloids. These cell number differences were not confined to ScxGFP fibroblasts (green) but extended to other populations (blue). In addition, ScxGFP fibroblasts only accumulated around the WT core in WT core // ScxGFP fibroblast assembloids, presumably either through increased core-directional migration or faster proliferation closer to the core. These observations are consistent with IL-6 being essential to increased cellularity and core (damage)-directed migration in this model (*Stauber et al., 2021*).

We went on to confirm these visual impressions using quantitative methods, finding a significantly increased total cell number in assembloids with a WT core compared to those with a KO core (*Figure 6B*, *Table 4*). The effect of IL-6 signaling on the proliferation of ScxGFP fibroblasts (*Figure 6C*, *Table 4*) was less pronounced compared to that on all populations, but the trend remained the same. To quantify migration, we analyzed the spatial distribution of ScxGFP fibroblasts by calculating the ratio between core-resident and extrinsic ScxGFP fibroblasts (*Figure 6D*, *Table 4*). The WT core // ScxGFP fibroblast assembloids exhibited the highest core-resident to extrinsic ScxGFP fibroblast ratio and KO core // ScxGFP fibroblast assembloids had a significantly lower core-resident to extrinsic ScxGFP fibroblast ratio. The cumulative spatial distribution of ScxGFP fibroblasts (*Figure 6E*, *Table 4*) supported these insights.

To further confirm the specific impact of IL-6 signaling on overall cell proliferation and core-directed migration of Scx⁺ fibroblasts, we desensitized the WT core // fibroblast assembloids to IL-6 by neutralizing IL-6R with tocilizumab and attempted to rescue the KO core // fibroblast assembloids by adding

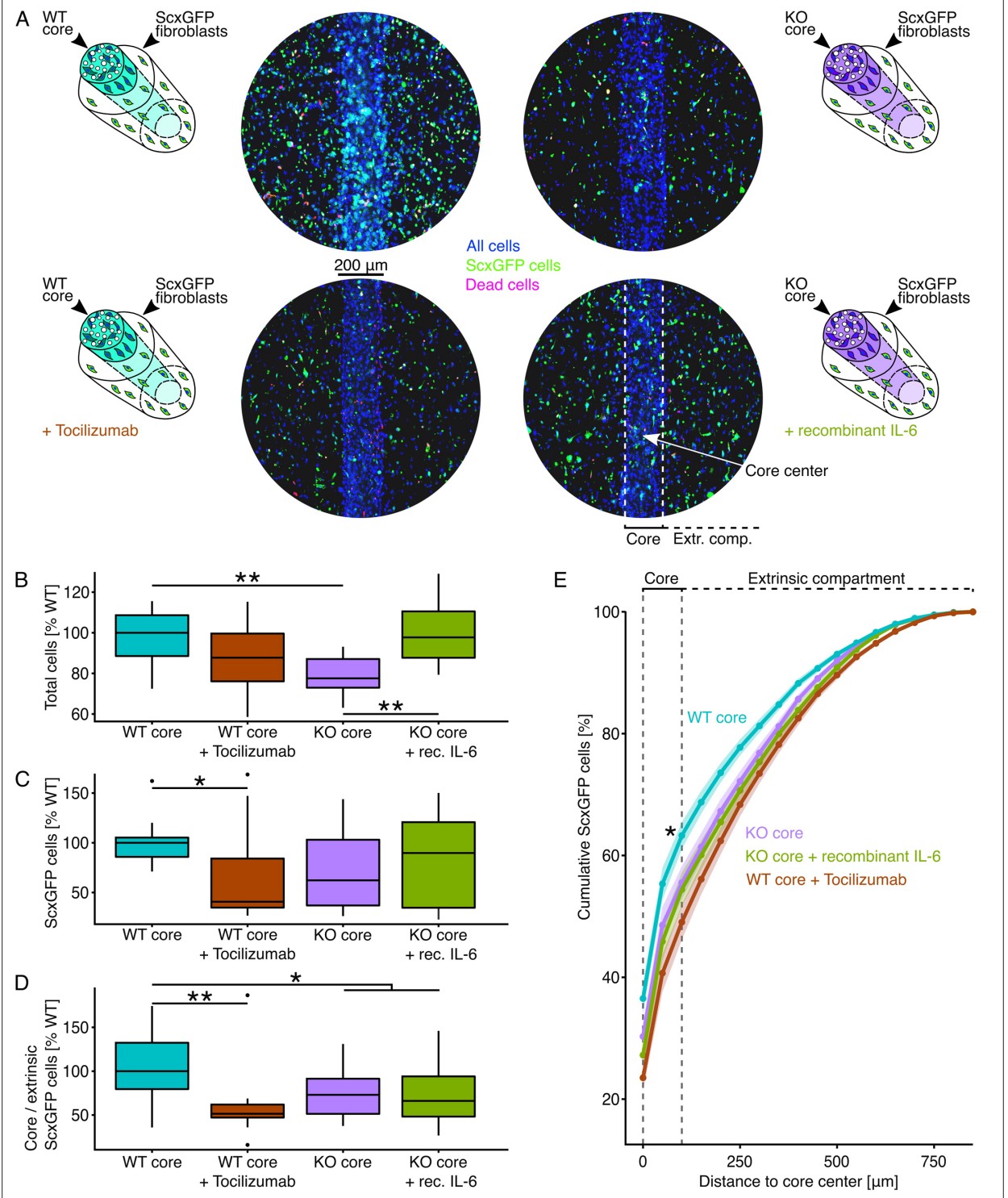

**Figure 6.** Cell proliferation around and ScxGFP fibroblast recruitment to core explants. (**A**) Illustrative depictions and representative fluorescence microscopy images of wildtype (WT) core explants surrounded by fibroblast populations from ScxGFP mice cultured with or without tocilizumab (10 µg/ml), and IL-6 knock-out (KO) explants cultured with or without recombinant IL-6 (25 ng/ml) for 7 days. All cells are colored in blue (NucBlue), ScxGFP fibroblasts in green (GFP), and dead cells in red (EthD). (**B, C, D**) Boxplots depicting the total number of cells, the number of ScxGFP cells, and the ratio between core-resident and extrinsic ScxGFP cells normalized to the WT median. Each datapoint was calculated from three representative fluorescence microscopy images taken from the same sample. The red datapoint matches the presented fluorescence microscopy image. The upper and lower hinges correspond to the first and third quartile (25th and 75th percentile), the middle one to the median, the whiskers extend from the hinges no further than 1.5 times the interquartile range, and the points beyond the whiskers are treated as outliers. (**E**) Lineplot depicting the cumulative

*Figure 6 continued on next page*

*Figure 6 continued*

percentage of ScxGFP cells depending on their distance from the center line of the core explant. The points and the line represent the mean cumulative percentages and the error bands the standard error of the mean (sem). The dashed line indicates locations inside the core area (biological replicates: N=12). Results of the statistical analysis are indicated as follows: *p<0.05, **p<0.01. The applied statistical test was the Mann-Whitney-Wilcoxon test, and lower hinges correspond to the first and third quartile (25th and 75th percentile), the middle one to the median, the whiskers extend from the hinges no further than 1.5 times the interquartile range, and the points beyond the whiskers are treated as outliers. (**E**) Lineplot depicting the cumulative percentage of ScxGFP cells depending on their distance from the center line of the core explant. The points and the line represent the mean cumulative percentages and the error bands the standard error of the mean (sem). The dashed line indicates locations inside the core area (biological replicates: N=12). Results of the statistical analysis are indicated as follows: *p<0.05, **p<0.01. The applied statistical test was the Mann-Whitney-Wilcoxon test.

recombinant IL-6 to compensate for their reduced IL-6 levels (*Figure 6A*, bottom panels). In alignment with the previous results and the hypothesis, IL-6 desensitization decreased the total cell number in trend (*Figure 6B*, *Table 4*), the ScxGFP cell number significantly (*Figure 6C*, *Table 4*), and the ratio between core-resident and extrinsic ScxGFP fibroblasts significantly as well (*Figure 6D*, *Table 4*). The addition of recombinant IL-6 to KO core // fibroblast assembloids significantly increased the total cell number and the number of ScxGFP cells, rescuing the WT phenotype of IL-6 enhanced cell proliferation in the extrinsic compartment. However, core-directed migration was not rescued by recombinant IL-6.

Fully in line with transcript signature changes detected in the extrinsic compartment, these data suggest that IL-6 signaling increased local cellularity in at least one of two ways. First, IL-6 stimulated both overall and specific ScxGFP cell proliferation. Second, IL-6 gradient effects (i.e. IL-6 induced secondary gradients) caused core-directed ScxGFP cell migration.

## Disrupting IL-6 signaling does not detectably alter Scx⁺ cell proliferation or recruitment into an acutely damaged Achilles tendon in vivo

After clarifying the role of IL-6 in activating fibroblasts in the assembloid model of chronic tendon disease, we sought to assess whether IL-6 signaling also enhances overall cell proliferation and migration of Scx⁺ fibroblasts to acute tendon damage. To test this, we first bred IL-6 WT and IL-6 KO mice with ScxGFP mice. Then, we assessed the presence of ScxGFP cells in the Achilles tendons of four IL-6 WT mice compared to those of four IL-6 KO mice 14 days after Achilles tenotomy. In addition, we used an 5-ethynyl-2-deoxyuridine (EdU) staining to assess the proliferation of cells within the healing tendon (*Figure 7A*).

Overall, the fluorescence microscopy images revealed a strong presence of ScxGFP cells (green) and EdU⁺ cells (magenta) in the neo-tendon (*Figure 7B*, tissue around the dashed circles) formed around the calcaneal Achilles tendon stumps (*Figure 7B*, AT within the dashed circles) after tenotomy (*Figure 7B*, left), but not in undamaged hindleg tendons (*Figure 7B*, middle). Similar levels of overall cellularity and presence of ScxGFP and EdU⁺ cells were observed in the calcaneal Achilles tendon stump and the surrounding neo-tendon of IL-6 WT and IL-6 KO mice. No observable trends or statistically detectable differences in the highly variable and complex ScxGFP cell migration patterns were detected between IL-6 WT and IL-6 KO mice (*Figure 7B*, left and right).

**Table 4.** Total cell numbers, ScxGFP cell numbers, and the ratios between core-resident and extrinsic ScxGFP cells in assembloids. The values were normalized to the wildtype (WT) median and are given as median(IQR).

| Condition | Total cell number norm. to WT (median(IQR)) | ScxGFP cell number norm. to WT (median(IQR)) | Core/extrinsic ScxGFP ratio, norm. to WT (median(IQR)) |
|---|---|---|---|
| WT core // ScxGFP fibroblasts | 100 (20.1)% | 100 (19.4)% | 100 (52.3)% |
| WT core // ScxGFP fibroblasts+tocilizumab | 87.7 (23.5)% | 40.7 (49.3)% | 51.5 (14.8)% |
| KO core // ScxGFP fibroblasts | 77.6 (14.1)% | 62.2 (66.1)% | 73(40)% |
| KO core // ScxGFP fibroblasts+IL-6 | 97.7 (22.8)% | 89.7 (86.1)% | 66.1 (45.9)% |

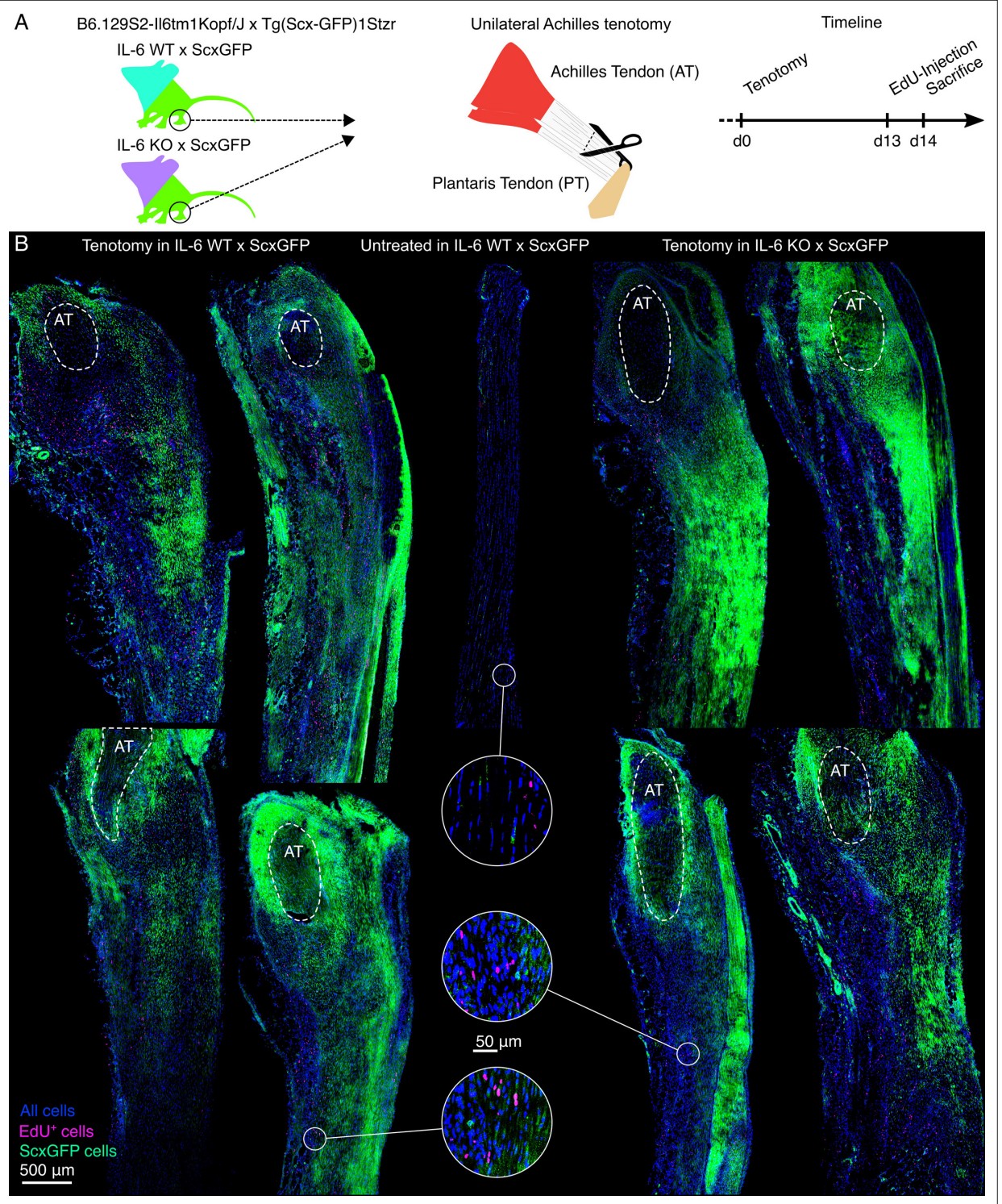

**Figure 7.** Cell proliferation around and Scx⁺ fibroblasts recruitment to acutely damaged mouse Achilles tendons 14 days after injury. (**A**) Illustrative depiction of the experimental setup and the time schedule. (**B**) Representative fluorescence microscopy images from all four mice assessed showing longitudinal mouse hindleg sections from IL-6 wildtype, ScxGFP (IL-6 WT x ScxGFP) Achilles tendons (AT) that underwent tenotomy (left), the contralateral untreated control (middle), as well as sections from IL-6 knock-out (IL-6 KO x ScxGFP) AT that underwent tenotomy (right). In addition to the signal provided by the ScxGFP cells (green), NucBlue was used to identify all cell nuclei (blue), and 5-ethynyl-2-deoxyuridine (EdU) was used to identify proliferating cells (magenta). The dashed circles indicate the remaining AT stump close to the calcaneus. The healing neo-tendon tissue surrounding the calcaneal AT stump bridges the gap to the AT stump connected to the calf muscles further down (not shown). In both compartments,

*Figure 7 continued on next page*

*Figure 7 continued*

ScxGFP cell distribution was highly variable across acutely damaged samples, with no observable trends or statistically detectable differences between the conditions (biological replicates: N=4).

The online version of this article includes the following figure supplement(s) for figure 7:

**Figure supplement 1.** Cell proliferation around and Scx+ fibroblast recruitment to damaged mouse Achilles tendons.

**Figure supplement 2.** CD146+ and TPPP3+ fibroblast recruitment to damaged mouse Achilles tendons.

## Activated, recruited, and proliferating extrinsic fibroblasts promote tendinopathy hallmarks in tendon core explants

Building upon the evidence that IL-6 potentiates tendon fibroblast activation and migration to damage in vitro, we then sought to clarify the nature of interactions between these recruited repair cells and the damaged tissue. We asked whether these activated fibroblasts might be capable of driving disease-relevant tissue processes. To assess this, we first looked at transcriptional changes induced in core explants when fibroblasts were present in the artificial extrinsic compartment by comparing them to explants cultured in an initially cell-free hydrogel (*Figure 8A*).

Exposing WT core explants to fibroblasts (WT core // fibroblasts) for 7 days increased 446 transcripts, decreased 217 transcripts, and left 19,694 transcripts unchanged (*Figure 8B and C*). In line with the previous paragraphs reporting fibroblast migration in vitro, some of the increased transcripts (i.e. Scx and Sox9) indicated an enrichment of Scx+ and/or Sox9+ fibroblasts in the WT core explants of WT core // fibroblast assembloids compared to those of WT core // cell-free assembloids. Similarly, GSEA on the full MSigDB cell-type signature gene sets proposed an amplified contribution of fibroblasts, fibroblast-like cells, and progenitor cells to the emerging assembloid phenotype (*Figure 8—figure supplement 1B*). In vivo, extrinsic (i.e. paratenon-derived) fibroblasts differentially express selected genes compared to tendon core (i.e. tendon proper-derived) fibroblasts (*Mienaltowski et al., 2019*). The GO gene sets annotated with these differentially expressed genes (DEGs) overlap with those enriched by DEGs between WT core // fibroblast and WT core // cell-free assembloids (*Figure 8—figure supplement 2A–D*). This could mean that the core explants exposed to extrinsic fibroblasts change into more paratenon-like tissue and again highlights the contribution of extrinsic fibroblast migration and accumulation to assembloid behavior.

To compare this phenotype to human tendinopathic tendons, we looked for ECM turnover-related transcripts that were enriched in human tendinopathic tendons compared to normal controls. Indeed, transcripts for *Col3a1*, *Col1a1*, *Mmp13*, *Mmp3*, and *Mmp9* were increased in core explants co-cultured with fibroblasts (*Figure 8*, *Table 5*). When combined through ORA, many of the top 30 biological processes enriched by significantly changed transcripts (adjusted p-value<0.01) were also enriched in human tendinopathic tendons (*Figure 1—figure supplement 2*). The curated list presented here (*Figure 8D*) pinpoints significantly enriched processes likely to be involved in tendinopathic hallmarks such as ECM turnover and tissue development, hypoxia and glucose metabolism, and hypercellularity.

Overall, it appears that extrinsic fibroblasts are sufficient to invoke several tendinopathic hallmarks in tendon core explants and accelerated catabolic matrix turnover in particular. We have previously reported an increase of IL-6 in the supernatant of WT core // fibroblast assembloids that correlated with an increased catabolic breakdown of the core (*Stauber et al., 2021*). The insights gained here connect this catabolic breakdown to gene sets involved in ECM remodeling. Another set of previously published experiments suggests that the ERK1/2 signaling cascade enriched in the core of WT core // fibroblast favors tissue breakdown as well (*Blache et al., 2021*; *Wunderli et al., 2020*).

## Disrupting IL-6 signaling in core explants diminishes emergence of tendinopathic hallmarks

So far, our results have shown that IL-6 signaling enhances proliferation and migration of fibroblasts toward the tendon core and that the presence of fibroblasts invokes tendinopathy-like changes in the tendon core in vitro. Consequently, the last step was to see whether an IL-6 KO not only prevents the fibroblast migration and proliferation, but also reduces fibroblast-invoked tendinopathic hallmarks in the core. To assess this, we again studied assembloids containing an IL-6 KO core, but this time focused on biological processes emerging in the core by leveraging bulk RNA-seq (*Figure 9A*).

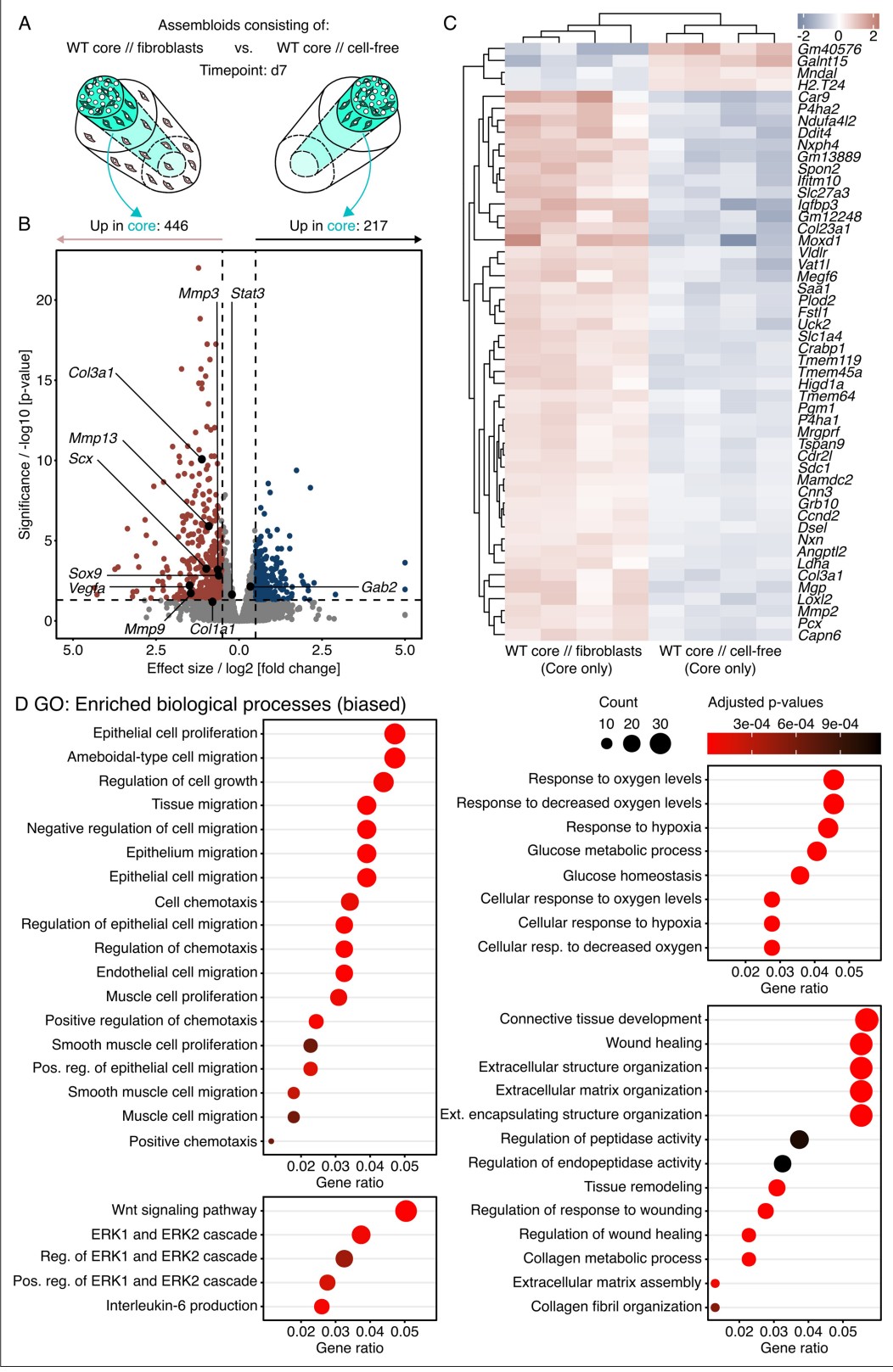

**Figure 8.** Transcript analysis of differentially regulated genes and pathways in wildtype (WT) core explants surrounded by a hydrogel seeded with fibroblasts compared to a WT core surrounded by a cell-free hydrogel. (**A**) Illustration depicting the assembloid combinations compared here (WT core // fibroblasts vs. WT core // cell-free), the assessed timepoint (**d7**), and the analyzed compartment (core only). (**B**) RNA-seq volcano plot of

*Figure 8 continued on next page*

*Figure 8 continued*

differentially expressed genes (DEGs). Genes colored in red have a log2 (fold change)>0.5, an adjusted p-value<0.05, and are considered to be significantly increased in the core of WT core // fibroblast assembloids. Genes colored in blue have a log2 (fold change)<–0.5, an adjusted p-value<0.05, and are considered to be significantly increased in the core of WT core // cell-free assembloids. The log2 and p-value thresholds are represented by the dashed lines. (**C**) Unsupervised hierarchical clustering of the top 50 DEGs. Genes are clustered by color with positive (red) or negative (blue) row-scaled z-scores. Columns represent individual samples (biological replicates: N=4). (**D**) Dotplots depicting a selection of gene ontology (GO) annotations significantly enriched (adjusted p-value<0.05) by the DEGs. The selection was biased by GO biological processes and gene set enrichment analysis (GSEA) hallmark annotations enriched in the human dataset (*Figure 1C and E*). The color of the circles represents their adjusted p-value, the size the number of enriched genes (count), and the x-axis the number of enriched genes in ratio to the total number of genes annotated to the gene set (gene ratio).

The online version of this article includes the following figure supplement(s) for figure 8:

**Figure supplement 1.** Detailed transcriptome analysis of genes up- and downregulated in wildtype (WT) core explants surrounded by a hydrogel seeded with extrinsic fibroblasts compared to a WT core surrounded by a cell-free hydrogel.

**Figure supplement 2.** Overlap between differentially expressed transcripts in in vitro assembloids and differentially expressed transcripts between Achilles tendon fibroblasts from the extrinsic compartment and the tendon core in vivo.

On the transcript level, we found 276 upregulated, 192 downregulated, and 20,204 unchanged genes in the core of KO core // fibroblast assembloids compared to that of WT core // fibroblast assembloids (*Figure 9B and C*). To see whether an IL-6 KO would partially reverse fibroblast-invoked hallmarks, we matched the list of DEGs (p-value <0.01) to the signatures in the GO database and then compared the surfacing enriched biological processes with those enriched by DEGs between the core of a WT core // fibroblast assembloid and a WT core // cell-free assembloid. The largest overlap lay in the signaling pathways (Wnt, ERK1/2, and IL-6), where 5/5 signatures for biological processes remained similarly enriched (*Figure 9E*). We found slightly fewer overlapping signatures connected to ECM turnover (6/14) and cellularity (6/18) while there seemed to be a disconnect in hypoxia and glucose metabolism (2/8). Overall, about a third of all signatures enriched by the presence of fibroblasts were also enriched by the IL-6 KO (*Figure 9E*). In contrast to the signatures emerging from the presence of fibroblasts however, the respective contribution of transcripts increased and decreased by the IL-6 KO to the enriched GO biological processes and molecular functions indicates a decreasing cellularity and ECM turnover in the core (*Figure 9—figure supplement 1*), which could mean that IL-6 signaling contributes to the gene expression behind these tendinopathy hallmarks.

As we have already demonstrated the tissue-level effects of IL-6 signaling on cellularity (*Figure 6*), we next wanted to examine the tissue-level consequences of changed transcript signatures for ECM turnover on core biomechanics. To do so, we measured the assembloid's linear elastic modulus as an indicator for their ability to resist longitudinal tension, which is one of the main functions of adult tendons (*Figure 9F*). In WT core // fibroblast assembloids, the linear elastic modulus decreased the most between the initial clamping of the core explant and 21 days of co-culture (light blue). The linear

**Table 5.** Effect sizes and p-values for selected transcripts.
The data describes differences in transcripts between the core explants from wildtype (WT) core // fibroblasts and those from WT core // cell-free assembloids.

| Transcript | Effect size | p-Value |
|---|---|---|
| *Col3a1* | 1.1 | 8.40E-11 |
| *Col1a1* | 0.8 | 0.06 |
| *Mmp13* | 0.91 | 1.20E-06 |
| *Mmp3* | 0.64 | 0.0006 |
| *Mmp9* | 1.45 | 0.019 |
| *Scx* | 0.98 | 0.00006 |
| *Sox9* | 0.6 | 0.001 |

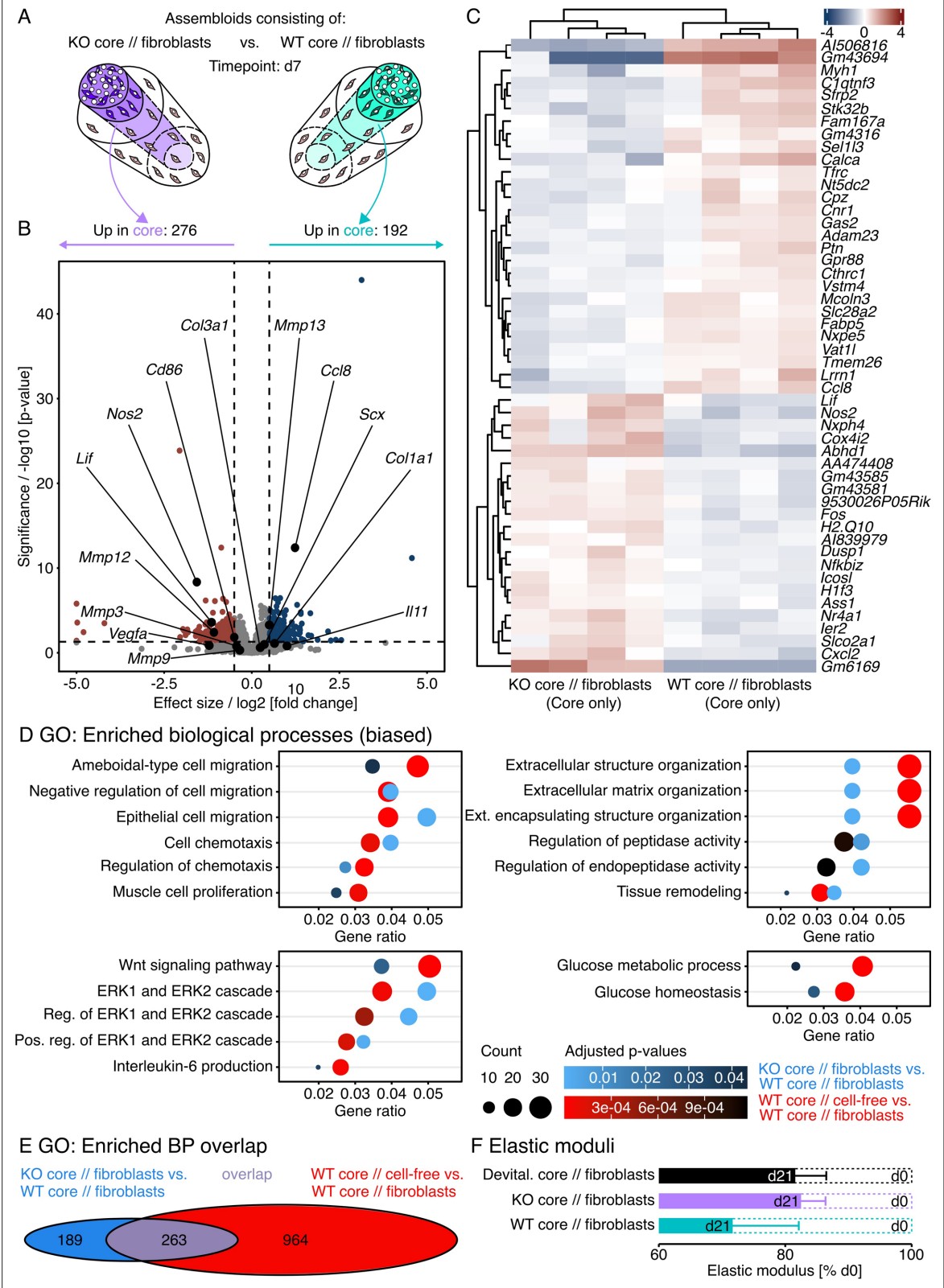

**Figure 9.** Transcript analysis of differentially regulated genes and pathways in IL-6 knock-out (KO) core explants surrounded by a hydrogel seeded with fibroblasts compared to a wildtype (WT) core surrounded by fibroblasts. (**A**) Illustration depicting the assembloid combinations compared here (KO core // fibroblasts vs. WT core // fibroblasts), the assessed timepoint (**d7**), and the analyzed compartment (core only). (**B**) RNA-seq volcano plot of differentially expressed genes (DEGs). Genes colored in red have a log2 (fold change)>0.5, a p-value<0.05, and are considered to be significantly

*Figure 9 continued on next page*

*Figure 9 continued*

increased in the core of KO core // fibroblast assembloids. Genes colored in blue have a log2 (fold change)<–0.5, a p-value<0.05, and are considered to be significantly increased in the core of WT core // fibroblast assembloids. The log2 and p-value thresholds are represented by the dashed lines. (**C**) Unsupervised hierarchical clustering of the top 50 DEGs. Genes are clustered by color with positive (red) or negative (blue) row-scaled z-scores. Columns represent individual samples (biological replicates: N=4). (**D**) Dotplots depicting a selection of gene ontology (GO) annotations significantly enriched (adjusted p-value<0.05) by the DEGs in both the WT core // cell-free vs. WT core // fibroblast assembloid comparison (red to black gradient) and the KO core // fibroblast vs. WT core // fibroblast assembloid comparison (light blue to black gradient). The selection was biased by enriched GO biological process and gene set enrichment analysis (GSEA) hallmark annotations in the human dataset (*Figure 1C and E*). The color gradient of the circles represents their adjusted p-value, the size the number of enriched genes (count), and the x-axis the number of enriched genes in ratio to the total number of genes annotated to the gene set (gene ratio). (**E**) Venn diagram depicting the number and the overlap (violet) of significantly enriched GO annotations for biological processes between the WT core // cell-free vs. WT core // fibroblast assembloid comparison (red) and the KO core // fibroblast vs. WT core // fibroblast assembloid comparison (blue). (**F**) Linear elastic moduli of devitalized (Devital.), IL-6 knock-out (KO), and wildtype (WT) core explants surrounded by hydrogel-embedded fibroblast populations at day 21 normalized to day 0 (biological replicates: N=8). The data are displayed as barplots with mean ± standard error of the mean (sem). The applied statistical test was the Mann-Whitney-Wilcoxon test and yielded no significant differences.

The online version of this article includes the following figure supplement(s) for figure 9:

**Figure supplement 1.** Detailed transcriptome analysis of genes up- and downregulated in knock-out (KO) core explants surrounded by a hydrogel seeded with fibroblasts compared to a wildtype (WT) core surrounded by a hydrogel seeded with fibroblasts.

elastic modulus of assembloids containing a KO core (violet) or a WT core explant devitalized through multiple freeze-thaw cycles (black) decreased as well, but not as fast or as strongly. In connection with the bulk RNA-seq data, the changes in the assembloid's ability to resist tension suggest that IL-6 signaling accelerates (catabolic) ECM turnover.

Normally, it would be hard to predict the effect of an accelerated catabolic ECM turnover in vitro on wound healing in vivo since both ECM degradation and synthesis are required to replace damaged tissue structures. However, previous studies have already reported a delayed wound healing response in IL-6 KO mice and our experiments here suggest that the decelerated catabolic ECM turnover could be responsible for this (*Lin et al., 2006*).

## Discussion

IL-6 is an attractive translational research target. Signaling cascades related to IL-6 are upregulated in tendon tissues after exercise, acute tissue damage, and in chronic tendon disease (*Nakama et al., 2006*; *Legerlotz et al., 2012*; *Langberg et al., 2002*). Little is currently understood of the precise role that IL-6 plays in these processes (*Arvind and Huang, 2021*). The goal of the present work was to clarify the role of IL-6 signaling in the tissues of non-sheathed tendons in the context of chronic damage, with a particular focus on inter-compartmental crosstalk between the damaged tendon core and extrinsic (reparative) fibroblasts populations targeted by it.

We first reanalyzed existing microarray data of (non-sheathed) tendinopathic tendons to verify a hypothesized (dys)functional role of IL-6 that seems to run partially via the JAK/STAT pathway enriched by transcripts increased in the tendinopathic samples. The JAK/STAT pathway interweaves with ERK1/2 downstream, which fits with recent data from our lab showing that ERK inhibition alone can prevent tendon matrix deterioration while reducing the secretion of IL-11, another member of the IL-6 cytokine family that was also upregulated in the tendinopathic tendons analyzed here (*Blache et al., 2021*; *Wunderli et al., 2020*). More generally, overactivation of JAK/STAT/ERK has been associated with autoimmune arthritis (*Ernst and Jenkins, 2004*; *Eulenfeld et al., 2012*).

While analysis of human samples indicated the expression of *IL6*, *IL6ST*, and *ADAM10* (known to transform membrane-bound IL-6R into soluble IL-6R) to be upregulated in tendinopathic tendons, the expression of *IL6RA* itself was downregulated. We speculate that this points toward increased trans-signaling aimed at stromal cells unable to express *IL6RA*, in this tendinopathic context particularly the stromal fibroblasts and fibroblast-like cell populations indicated to be enriched by the cell-type signatures (*Nowell et al., 2003*; *Rose-John, 2012*). This theory is supported by GO terms enriched in tendinopathic samples compared to controls relating to increased morphogenesis and wound healing, both processes supposedly powered by (reparative) fibroblasts (*Dyment et al., 2014*; *Niu et al., 2020*). Most of the remaining top 20 enriched GO terms pointed toward increased cell proliferation, potentially increasing the leverage of the proliferating fibroblasts. Indeed, staining human samples

with CD90, an established marker of reparative fibroblasts typically present in healing tissues, *Ho et al., 2019*; *Li et al., 2021*, confirmed a higher percentage of CD90+ cells in (non-sheathed) tendinopathic compared to normal control tendons.

While beneficial in controlled dosages during normal wound healing, both excessive hypercellularity and imbalanced wound healing are hallmarks of tendinopathy (*Li et al., 2007*; *Millar et al., 2021*; *Sharma and Maffulli, 2006*; *Harvey et al., 2019*). To decipher their connections to IL-6 signaling, we investigated IL-6/IL-6R concentration gradients, sources, and targets in patient-derived tissue sections. In tendinopathic patients, the presence of IL-6 over the whole tissue was smaller than expected. However, while IL-6 seemed to be largely confined to the extrinsic compartment in normal control tendons, it was distributed across compartments in tendinopathic tendons. The resulting gradient differences could play a role in the emergence of disease hallmarks (*Crowe et al., 2023*). In contrast to the relative transcript-level decrease of *IL6RA* in tendinopathic tendons indicated by the microarray, its presence was increased on the protein level. Although we can only speculate on the clearance rate of both soluble and membrane-bound IL-6R as well as of the cells carrying them, the mismatch in its presence on the gene transcription and the protein expression levels could be a legacy from earlier disease stages.

To identify sources and targets of IL-6 and IL-6R, we co-stained the tendinopathic tissue sections with established markers for macrophages (CD68) and reparative fibroblasts (CD90). Presuming that IL-6 is more concentrated around its sources than its targets due to diffusion, it seems that although both CD68+ and CD90+ cells express IL-6, the contribution of CD90+ cells is significantly larger. This observation is in line with sources of IL-6 identified in growing mouse tendons (*Bautista et al., 2023*). The number of CD90+ cells among IL-6R+ cells was also significantly larger than that of CD68+ cells. Since it has yet to be shown that stromal fibroblasts express and translate IL-6R, a better explanation might be that the IL-6R on CD90+ cells were originally produced by, i.e., the CD68+ cells and ended up on CD90+ cells as part of a trans-signaling process. Why and how CD90+ cells are targeted by soluble IL-6R as part of IL-6 trans-signaling could be an interesting part of future research.

To dissect the specific effect of IL-6 signaling on reparative fibroblasts, we turned to tendon assembloids: hybrid explant // hydrogel models of core tendon damage and repair that were recently developed in our lab and which identified an underloaded core explant as a potentially biologically relevant source of IL-6 (*Stauber et al., 2021*). The simultaneously increased and imbalanced matrix tissue turnover perpetuated by the crosstalk between the core and an extrinsic fibroblast population seemed to put the system into a prime position to decipher the connections between IL-6, hypercellularity, and (catabolic) matrix turnover. Furthermore, their compartmentalized design adequately mimics the structure of non-sheathed tendons.

Replacing the WT core explant with an IL-6 KO core explant in our assembloids was sufficient to reduce the expression of genes in gene sets related to matrix turnover, proteolysis, cell proliferation, and cell migration in the extrinsic fibroblast population. We confirmed the IL-6-induced gene-level differences regarding cell migration by exploiting trackable ScxGFP fibroblasts and demonstrated effective manipulation of both recruitment and proliferation of ScxGFP cells through IL-6. We did so by using the IL-6 inhibitor tocilizumab to desensitize resident cell populations to IL-6 and recombinant IL-6 to replace the IL-6 not secreted by an IL-6 KO core. Recombinant IL-6 rescuing proliferation but not migration of ScxGFP cells highlights the necessity of an IL-6 gradient and/or the establishment of a secondary cytokine gradient (e.g. TGF-β) by IL-6 (*Tan et al., 2021*). Alternatively, IL-6 has also been described as an energy allocator in other musculoskeletal tissues and could in this function be accelerating a diverse range of processes (i.e. migration) by increasing the baseline cell metabolism (*Kistner et al., 2022*). Fittingly, the extrinsic cell populations around an IL-6 KO core upregulated biological processes related to oxidative stress and hypoxia, which indicates a disrupted energy allocation. Since we found similar biological processes to be positively enriched in human tendinopathic tendons compared to controls, future studies should more closely examine the role of hypoxic signaling in the pathogenesis of tendinopathy.

To transfer these insights from our in vitro experiments to an in vivo setup, we used an Achilles tenotomy model of the in vivo tendon damage response. Although the number of Scx+ fibroblasts did increase in injured tendons, the in vivo experiments did not detect an effect of IL-6 on activating Scx+ fibroblasts (*Korcari et al., 2022*) and making them migrate into or proliferate faster in the damaged and unloaded Achilles tendon stump. In addition, previous studies with mice subjected to a patellar

punch procedure reported only marginally reduced mechanical properties in the healing patellar tendon of IL-6 KO mice compared to the WT after an acute injury (*Lin et al., 2006*). However, we are cautious in interpreting these in vivo results, mainly because 14 days after an Achilles tenotomy, the tendon niche condition represents an acute rather than a chronic tendon lesion (*Jones et al., 2006*; *Sugg, 2014*). Also, even with breeding a novel IL-6 KO x ScxGFP mouse line to alleviate widely known problems of Scx antibodies, the collagen matrix still caused considerable background noise in our images (*Lin et al., 2006*). Furthermore, the ablation of IL-6 might have led to the elevation of compensatory IL-6 superfamily ligands (or other attractant chemokines or cytokines) and one can currently only speculate on their in vivo distribution as well as the resulting patterns of fibroblast migration (*Dyment et al., 2013*; *Langberg et al., 2002*). To confirm this hypothesis, future in vivo studies could deploy IL-6 inhibitors targeting other members of the IL-6 superfamily as well.

Since the migrating Scx⁺ fibroblasts in vitro were apparently targeting the damaged core tissue (likely to support the limited intrinsic regenerative potential of the explanted tendon core secreting IL-6 in the first place) (*Stauber et al., 2020*), the next set of experiments we conducted in this work focused on the effects of the activated and recruited fibroblasts on the tendon core.

According to existing literature and results presented here, Scx⁺ fibroblasts are increasingly present in in vivo murine adult tendon lesions (*Dyment et al., 2013*; *Tan et al., 2021*) and depleting them alternately improves or impairs adult tendon healing depending on the timepoint of depletion (*Korcari et al., 2022*; *Best et al., 2021*). One underlying reason could be that adult, Scx⁺ fibroblasts hold a bi-fated potential that enables a cartilage-like differentiation when exposed to mechanical compression or tensile unloading (*Howell et al., 2017*; *Kult et al., 2021*). Our in vitro assembloid model captured these behaviors as well with increased Scx/Sox9 transcripts and enriched gene sets indicating a stronger presence of fibroblasts alongside cartilage development in a core surrounded by (migrating) fibroblasts compared to one embedded in a cell-free hydrogel. Besides, genes differentially expressed in a core explant surrounded by fibroblasts enriched gene sets related to hypercellularity, ERK1/2 signaling, oxygen/glucose metabolism, and ECM turnover. Most of these processes were reduced in an IL-6 KO core, speculatively because of the reduced presence of extrinsic fibroblasts resulting from the reduced migration and proliferation. On the tissue level, we also found signs for a decreased catabolic matrix turnover in the more stable mechanical properties of IL-6 KO core explants, a process likely linked to ERK1/2 signaling (*Blache et al., 2021*).

In summary, our data consistently point to IL-6 signaling targeting reparative fibroblasts being upregulated in chronic human (non-sheathed) tendon lesions in a manner that directly leads to fibroblast recruitment and proliferation as well as aberrant morphogenesis/matrix turnover. This activity contributes to typical hallmarks of tendinopathy including hypercellularity and loss of biomechanical tissue integrity.

## Materials and methods

### Human microarray data analysis

We reanalyzed a microarray dataset (GEO: GSE26051) from 2011 with contemporary methods (principal component analysis, volcano plots, heatmaps, GSEA, and ORA), focusing on the IL-6 signaling cascade. All steps from downloading the dataset to the differential expression computation were conducted in RStudio ('Prairie Trillium', https://github.com/rstudio/rstudio; *rstudio, 2022*) running R version 4.1.2. Overall, we closely followed the steps described here: https://sbc.shef.ac.uk/geo_tutorial/tutorial.nb.html (last visited: May 2, 2022). First, we log2-transformed the expression values and checked their distribution with boxplots. Since the original dataset was gathered from a wide variety of anatomical locations and differently aged patients (*Supplementary file 1*), we started with a principal component analysis to filter out outliers. Based on the clustering, we identified differences between sheathed and non-sheathed tendons (*Figure 1—figure supplement 2*). For the following analysis, we therefore excluded samples gathered from sheathed tendons:

GSM639749 (EDC), GSM639751 (Flexor-Pronator), GSM639756 (Flexor-Pronator), GSM639761 (ECRB), GSM639765 (ECRL), GSM639772 (ECRB), GSM639774 (Flexor-Pronator), GSM639779 (Flexor-Pronator), GSM639784 (ECRB), GSM639788 (ECRB).

To improve the power to detect DEGs, we filtered out genes with very low expression. We considered 50% of genes to not be expressed and therefore used the median expression as the cut-off.

In addition, we only kept genes expressed in more than two samples for further analysis and calculated the average of replicated probes. Afterward, we applied the empirical Bayes' step to receive the differential expression values and p-values. We plotted the DEGs as a volcano plot and annotated IL-6 signaling-related genes, other cytokines of the IL-6 family, their respective receptors, and genes involved in matrix turnover. Here, we considered genes with a p-value<0.05 to be differentially expressed. In addition, we plotted the row scaled z-scores of a selection of the annotated genes in a heatmap.

To produce the GO annotations, we fed the list of IDs from DEGs into the enrichGO function from the clusterProfiler package (version 3.0.4, here, last visited: October 31, 2022) using the org.Hs.eg.db as reference and the Benjamini-Hochberg method to calculate the false discovery rate (FDR)/adjust the p-values. To visualize the data, we used the dotplot function from the enrichPlot package (https://rdrr.io/bioc/enrichplot/, last visited: October 31, 2022). We also looked at the increased (logFC>0) and decreased (logFC<0) transcripts in isolation to estimate their contribution to the enrichment and give it directionality.

For the GSEA performed in RStudio with clusterProfiler, we used the human hallmark and the cell-type signature gene set annotations from the molecular signature database (MSigDB, https://www.gsea-msigdb.org/gsea/msigdb/, last visited: October 31, 2022) after ranking the genes according to their p-value. We used the following input parameters: pvalueCutoff = 1.00, minGSSize = 15, maxGSSize = 500, and eps = 0. Lastly, we used the gseaplot and dotplot functions from the enrichPlot package to plot the data and the sign of the enrichment score/NES to estimate the directionality. The exact code can be found in the supplementary material (**Source code 1**).

## Human immunohistological stainings

Tendon tissues from tendinopathic and normal control tendons were collected with informed consent including consent to publish from human patients undergoing treatment at the University Hospital Balgrist (ethical permission numbers 2016-02665 and 2020-0119 as approved by the institutional review board of the Canton of Zurich). Patient data and images are depicted in **Figure 2—figure supplement 1**. We cut transversal cryosections (10 µm thickness) using a low-profile microtome blade (DB80 LX, BioSys Laboratories), collected them on a glass slide, and let them dry for 1 hr before storing them at –80°C until further use. Prior to staining, sections were air-dried for 30 min at RT (room temperature) and washed 3× with PBS for 5 min each. Then, sections were permeabilized and blocked with 3% BSA (bovine serum albumin) in PBS-T (PBS+0.1% Triton X) for 1 hr at RT. We washed the sections again, added the primary antibody for CD90 (Abcam, ab181469, diluted 1:100 for the co-staining with IL-6R and GeneTex, GTX130072 diluted 1:200 for the co-staining with IL-6), CD68 (Abcam, ab955, diluted 1:50), IL-6 (R&D Systems, MAB2061R, diluted 1:200), and IL-6R (Absolute Antibody, ab00737-23.0, diluted 1:100) in PBS-T with 1% BSA. We covered them with parafilm and left them overnight in a humid chamber at 4°C. Afterward, we washed them again (3×5 min with PBS) before adding the matching secondary antibodies (diluted 1:200 in PBS with 1% BSA) to the samples as well as the secondary antibody controls.

The sections were then washed again (3×5 min with PBS+1×5 min with ultra-pure water) before mounting the coverslip with ROTIMount FluorCare DAPI (Roth). We used the Leica SP8 automated inverse confocal laser scanning microscope for acquiring the images, which we then processed with ImageJ 1.53q and RStudio (**Source code 2** and **Source code 3**).

## Mouse tissue harvest

All animal experiments were approved by the responsible authorities (Canton Zurich license number ZH104-18 and ZH058-21).

We extracted tail tendon core explants and Achilles tendons from 12- to 15-week-old male and female Tgf(Scx-GFP)1Stzr and B6.129S2-Il6tm1Kopf/J mice (knock-out: KO, wildtype: WT) as described previously (**Figure 4B and C**; **Stauber et al., 2021**; **Stauber et al., 2024**). We isolated the core explants from the tail and only kept those with a mean diameter between 100 and 150 µm in standard culture medium (DMEM/F12 GlutaMAX with 10% fetal bovine serum, 1% penicillin/streptomycin, 1% amphotericin, 200 µM L-ascorbic acid) until clamping them. Meanwhile, we separated the Achilles tendon from the calcaneus and the calf muscle using a scalpel and washed them with PBS before starting the digestion process (Standard culture medium without L-ascorbic acid but 2 mg/ml

collagenase for 24 hr at 37°C). After digestion, we cultured the cells on 2D tissue culture plastic in standard culture medium and used the resulting mixed fibroblast population between passage 2 and 4 (*Figure 4—figure supplement 1*, *Figure 4—figure supplement 2*). All medium components were purchased from Sigma-Aldrich, except for the ascorbic acid (Wako Chemicals) and the collagenase (Thermo Fisher).

## Collagen isolation

We isolated collagen-1 from rat tail tendon fascicles following an established protocol (*Rajan et al., 2006*). Briefly, tendon explants were extracted from the tail of adult (>8 weeks) female Sprague-Dawley rats with surgical clamps. Then, the collagen was dissolved by sequentially putting the core explants into acetone (5 min), 70% isopropanol (5 min), and finally 0.02 N acetic acid (48 hr). The resulting viscous solution was mixed in a house-ware blender and then frozen at –20°C. Lyophilization at –20°C turned the viscous solution into a dry collagen sponge, which was stored at –80°C and aliquots thawed when needed. Upon thawing, the collagen aliquot was mixed with 0.02 N acetic acid and then centrifuged (15,000 rpm for 45 min) at 4°C. The supernatant was then sterilized with SPECTRAPOR dialysis bags first in non-sterile acetic acid (1 hr), then 1% chloroform in ddH$_2$O (1 hr), and finally sterile acetic acid (three times for 2 days each). The concentration of the resulting solution was determined with a hydroxyproline assay (Sigma-Aldrich, MAK008), the purity was assessed with SDS-PAGE and western blots, and the solution itself was stored at 4°C until usage in the experiments.

## Hydrogel preparation, core explant embedding, and assembloid culture

As described previously (*Stauber et al., 2021*; *Stauber et al., 2024*), core explants were fixated with clamps, placed into molds lining silicone chambers, and tensioned. These molds were then filled with cell-free or extrinsic fibroblast-laden collagen hydrogels. One hydrogel consisted of 10 μl PBS (20×), 1.28 μl of 1 M NaOH (125×), 8.72 μl double-distilled water (ddH$_2$O, 23×), 80 μl collagen-1 (2.5× or 1.6 mg/ml) and 100 μl culture media or cell suspension (2×). All hydrogel components were kept on ice to prevent pre-mature crosslinking. Co-culture medium (DMEM/F12 high glucose, 10% FBS, 1% non-essential amino acids, 1% penicillin/streptomycin, 1% amphotericin, 200 μM L-ascorbic acid, 20 ng/ml macrophage-colony stimulating factor) was added to stable hydrogels after 50 min of polymerization at 37°C and tension was released. The assembloids were then cultured under tendinopathic niche conditions (37°C, 20% O$_2$) with two media changes per week until the determined timepoint (*Wunderli et al., 2020*). We used a final concentration of 25 ng/ml recombinant IL-6 (PeproTech, 216-16) in those assembloids to be stimulated by it, and a final concentration of 10 μg/ml tocilizumab (TargetMol, T9911) in those assembloids to be inhibited by it.

## RNA isolation for genome-wide RNA-seq (bulk RNA-seq)

We pooled 20–24× 2 cm core explants and 2 extrinsic fibroblast-laden collagen hydrogels separate from each other and snap-froze them in liquid nitrogen. The core explant pools were generated from a single mouse and represent one biological replicate each. The collagen hydrogel pools contained a mixed population comprising migratory cells from the embedded core (same mouse) and the initially seeded mixed fibroblast population (cells pooled from six mice). The frozen samples were pulverized by cryogenic grinding (FreezerMill 6870, SPEX SamplePrep) and further processed with the RNeasy micro kit (QIAGEN) according to the manufacturer's instructions. We used the NanoDrop 1000 spectrophotometer 3.7.1 (Thermo Fisher) to measure RNA concentration and purity, and the 4200 TapeStation System (Agilent) to measure RNA quality. For each condition (WT core // cell-free, WT core // fibroblasts, KO core // fibroblasts), all six of the collagen hydrogels pools but only four of the core explant pools passed both integrity control (RIN≥2) and had a sufficiently high RNA concentration (30–100 ng/μl) for genome-wide RNA-seq.

We submitted those pools to the functional genomics center Zurich (https://fgcz.ch/, last visited May 6, 2022) for the Illumina (NovaSeq 6000) TruSeq TotalRNA stranded sequencing protocol including library construction from total RNA using ribo-depletion, library QC, sequencing, and data delivery.

## RNA-seq data processing and bioinformatic analysis

We used the R-based SUSHI framework of the Functional Genomics Center Zurich (ETH Zurich and University of Zurich) to perform primary level bioinformatics. Specifically, we used the FastqcApp, the

FastqScreenApp, and the RnaBamStatsApp for quality control, the KallistoApp (sleuth) to calculate transcript abundance after pseudoalignment, the CountQCApp to quality control after counting reads, and the DESeq2App for differential expression analysis. We then used the shiny toolset developed by the Functional Genomics Center Zurich (https://github.com/fgcz/bfabricShiny, copy archived at *Functional Genomics Center Zurich ETHZ | UZH, 2022*, last visited May 6, 2022) based on b-fabric and R to generate the annotated volcano plots, heatmaps, and gene set functional enrichment by applying the hypergeometric ORA with the following settings:

## Volcano plot

| Comparison | p-Value | p-Value threshold | Log2FC threshold |
|---|---|---|---|
| Core of WT core // cell-free – Core of WT core // fibroblasts | FDR-adjusted | 0.05 | 0.5 |
| Core of WT core // fibroblasts – Core of KO core // fibroblasts | Raw | 0.05 | 0.5 |
| Fibroblasts from WT core // fibroblasts – Fibroblasts from KO core // fibroblasts | Raw | 0.05 | 0.5 |

## Heatmap

| Comparison | #Genes | Scale | Count method |
|---|---|---|---|
| WT core // cell-free – WT core // fibroblasts | Top 50 up and down | Diverging | Normalized+Log2 |
| Core of WT core // fibroblasts – Core of KO core // fibroblasts | Top 50 up and down | Diverging | Normalized+Log2 |
| Fibroblasts from WT core // fibroblasts – Fibroblasts from KO core // fibroblasts | Top 50 up and down | Diverging | Normalized+Log2 |

## Overrepresentation analysis

| Comparison | Input p-value | Output p-value |
|---|---|---|
| WT core // cell-free – WT core // fibroblasts | FDR-adjusted<0.01 | FDR-adjusted<0.05 |
| Core of WT core // fibroblasts – Core of KO core // fibroblasts | Raw<0.01 | FDR-adjusted<0.05 |
| Fibroblasts from WT core // fibroblasts – Fibroblasts from KO core // fibroblasts | Raw<0.01 | FDR-adjusted<0.05 |

We also looked at the increased (logFC>0) and decreased (logFC<0) transcripts in isolation to estimate their contribution to the enrichment and give it directionality. The emapplot of enriched biological processes independent of increased and decreased transcripts was generated in RStudio using the enrichPlot package.

We again used RStudio and the clusterProfiler package to perform the GSEA, taking the mouse hallmark and the cell-type signature gene set annotations from the molecular signature database (MSigDB, https://www.gsea-msigdb.org/gsea/msigdb/, last visited: October 31, 2022) as reference after ranking the genes according to their signed log2 ratio. We used the following input parameters: pvalueCutoff = 1.00, minGSSize = 15, maxGSSize = 500, and eps = 0. Lastly, we used the gseaplot and dotplot functions from the enrichPlot package to plot the data and the sign of the enrichment score/ NES to estimate the directionality.

The RNA-seq data gathered from assembloids as discussed in this publication have been deposited in NCBI's Gene Expression Omnibus (*Edgar et al., 2002*) and are accessible through GEO series accession number GSE214015 (https://www.ncbi.nlm.nih.gov/geo/query/acc.cgi?acc=GSE214015).

The in vivo mouse RNA-seq data comparing paratenon-derived to tendon proper-derived fibroblasts have previously been published in an open-access database (PRJNA399554) (*Mienaltowski et al., 2013*). To reanalyze this data, we used the same tools and parameters as for the assembloid

analysis (GO input: FDR-adjusted p-value<0.01). The overlapping DEGs and GO terms were calculated and the resulting Venn diagrams plotted with basic RStudio functions (i.e. intersect & draw. pairwise.venn).

## Quantifying total cell proliferation, ScxGFP cell proliferation, and ScxGFP cell recruitment to WT and KO core explants

In the assembloids used here, core explants from WT and homozygous IL-6 KO B6.129S2-Il6tm1Kopf/J mice were embedded with ScxGFP fibroblasts from homo- and heterozygous Tgf(Scx-GFP)1Stzr mice. After 7 days, the assembloids were removed from the clamps and washed with PBS before staining them with ethidium homodimer (EthD-1, Sigma-Aldrich, 2 mM stock in DMSO) diluted to 4 µM with PBS (20 min, 37°C). They were then again washed with PBS, fixated with 4% formaldehyde (Roti-Histofix, Karlsruhe) for 1 hr at RT, washed again with PBS, and stored in PBS at 4°C. Immediately before the imaging, nuclei were stained with NucBlue Live Ready Probes Reagent (R37605, Thermo Fisher) for 1 hr at RT. We used the Nikon Eclipse T*i*2 confocal scanning microscope controlled by NIS-Elements to acquire the images (three per sample), which we then processed with ImageJ 1.53q. Briefly, we first registered all cell locations by creating a mask from the NucBlue channel. Then, we put this mask over the ScxGFP channel and measured the fluorescence intensity at the identified cell locations. We then transferred the signal intensity per location data to RStudio, where we first calculated the total cell numbers of all the images of one sample combined and normalized them to the WT median. Afterward, we determined the fluorescence threshold for the ScxGFP-signal (using density plots and a negative control image) and applied this threshold to the dataset. We then calculated the total ScxGFP cell numbers for each sample and normalized them to the WT median. Finally, we combined the cell location with the fluorescence intensity data to find the distance from the core where most of the ScxGFP cells were located and to calculate the ratio between ScxGFP present at the core and those present in the surrounding extrinsic hydrogel (*Source code 4*, *Source code 5*).

## Quantifying mechanical properties of assembloids

We mounted the assembloids to a custom-made uniaxial stretching device equipped with a load cell as described previously (*Stauber et al., 2021*). After five cycles of pre-conditioning to 1% $L_0$, the assembloids were then stretched up to 2% $L_0$ to measure the linear elastic modulus (Emod) with a pre-load of 0.03 N. This measurement was repeated after 21 days (d21) of culture. We used MATLAB R2017a and RStudio to read out the linear elastic modulus and normalize it to the measurement immediately after assembloid fabrication (d0). Media was changed every 2–3 days. For the corresponding condition, the core explants were devitalized by snap-freezing them repeatedly in liquid nitrogen.

## Achilles tenotomy

Adult WT and homozygous IL-6 KO *B6.129S2-Il6tm1Kopf/J* (*Kopf et al., 1994*) x Tg(Scx-GFP)1Stzr mice (between 12 and 15 weeks of age) of both genders were anesthetized by isoflurane inhalation. While the mice were anesthetized, we transected the Achilles tendon of the right hindlimb by creating a small incision in the tendon midsubstance (*Figure 7A*). The contralateral hindlimb was used as the undamaged control. After the surgical intervention, we closed the skin wound with an 8/O prolene suture (Ethicon, W8703) and administered an analgesic (buprenorphine, 0.1 mg/kg s.c., 26 G needle). At 1 week (*Figure 7—figure supplements 1 and 2*) and 2 weeks (*Figure 7*) post-tenotomy, we injected 10 µl/g of EdU into each mouse of the 3 weeks group and euthanized them 24 hr later with $CO_2$. We collected the plantaris and Achilles tendon/neotendon from both hindlegs for histology. The isolated tissues were placed in OCT (TissueTek), cooled down on dry ice, and then stored at –80°C until further use.

## Immunofluorescence microscopy of mouse Achilles tendon sections

We cut transversal (1 week) and longitudinal (3 weeks) cryosections (10 µm thickness) using a low-profile microtome blade (DB80 LX, BioSys Laboratories), collected them on a glass slide, and let them dry for 1 hr before storing them at –80°C until further use. Prior to staining, sections were air-dried for 30 min at RT and washed 3× with PBS for 5 min each. Then, sections were permeabilized and blocked with 3% BSA in PBS-T (PBS+0.1% Triton X) for 1 hr at RT. We then washed the sections again and incubated the sections that were previously stained with EdU with a reaction cocktail (Jena Bioscience,

CLK-074, CuAAC Cell Reaction Buffer Kit [THPTA based]) prepared according to the manufacturer's instructions (440 µl reaction buffer, 10 µl CuSO$_4$, 1 µl (2 µM) Alexa Fluor azide 647, and 50 µl reducing agent) at RT for 45 min. We washed the sections again, added the primary antibody for Scx (abcam, ab58655, diluted 1:200 in PBS-T with 1% BSA), TPPP3 (Invitrogen, PA5-24925, 1:200), or CD146 (BIOSS, bs-1618R, 1:200) to the sections from the 1-week timepoint, and a GFP-antibody (Abcam, ab290, 1:500 in PBS-T with 1% BSA) to those from the 3-week timepoint. We covered all the sections with parafilm and left them overnight in a humid chamber at 4°C. Afterward, we washed them again (3×5 min with PBS) before adding the matching secondary antibodies (diluted 1:200 in PBS with 1% BSA) to the samples as well as the secondary antibody controls.

The sections were then washed again (3×5 min with PBS+1×5 min with ultra-pure water) before mounting the coverslip with ROTIMount FluorCare DAPI (Roth). We used the Nikon Eclipse T*i*2 confocal scanning microscope controlled by NIS-Elements for acquiring the images, which we then processed with ImageJ 1.53q and RStudio as described previously (see 'Quantifying total cell prolifer-ation, ScxGFP cell proliferation, and ScxGFP cell recruitment to WT and KO core explants'). To quan-tify the migration through the location of Scx$^+$/TPPP$^+$/CD146$^+$ cells in *Figure 7—figure supplements 1 and 2*, we defined the lesional Achilles tendon area as a circle with a 480 µm radius set in the center of the Achilles tendon.

The researcher performing the staining, imaging, and analysis of the Achilles tendon sections was blinded to the conditions by marking the samples with numbers only.

## Secretome analysis

Culture medium was enriched with the secretome of the different assembloids (WT core // cell-free, KO core // cell-free, WT fibroblasts in a hydrogel) for 3 days and until day 7 of the assembloid/ hydrogel culture. IL-6 was quantified using a custom-made multiplex U-PLEX for mouse biomarkers (Meso Scale Discovery) according to the manufacturer's instruction. Plates were read with the MESO Quickplex SQ120 (Meso Scale Discovery) and analyzed with Discovery Workbench 4.0.13 (https:// www.mesoscale.com/en/products_and_services/software). The plate was read with the Epoch Micro-plate Spectrophotometer (Biotek), and the data were analyzed with RStudio.

## Statistical analysis and graph design

Data curation, statistical analysis, and plotting was done in RStudio ('Prairie Trillium', 9f796939, February 16, 2022, https://github.com/rstudio/rstudio, copy archived at *rstudio, 2022*) running R version 4.1.2. For normally distributed datasets, statistical information was obtained by ANOVA followed by Tukey's post hoc tests for pairwise comparisons. Else, the non-parametric Wilcoxon rank sum test was applied, directionally matching the data (less, greater, two-sided). For all tests, we tested the level of p-values. The mean and the standard error of the mean (sem) were reported for the following data: cumulative percentages of ScxGFP fibroblasts in assembloids, elastic modulus of the assembloids, and cumu-lative percentages of Scx$^+$/TPPP$^+$/CD146$^+$ cells in the in vivo tenotomy model. We used bar and/or point plots to depict the mean and error bars/bands to depict the sem. We reported the median and interquartile range (IQR) in assembloids for the total cell number, the number of ScxGFP cells, and the ratio between core-resident and extrinsic ScxGFP cells, as well as for the total cell number, the number of Scx$^+$ cells, and the ratio between Achilles and neotendon-resident Scx$^+$/ TPPP$^+$/CD146$^+$ cells in the in vivo tenotomy. These values were depicted as boxplots with the upper and lower hinges corresponding to the first and third quartile (25th and 75th percentile), the middle one to the median, the whiskers extending from the upper/lower hinge to the largest/smallest value no further than 1.5 times the IQR, and dots representing data beyond the whiskers. Results of the statistical analysis are indicated as follows: *p<0.05, **p<0.01, ***p<0.01.

The open-source graphics software Inkscape 0.92.3 (https://inkscape.org/, last visited May 9, 2022) was used to finalize the graph design.

## Acknowledgements

This work was funded by the ETH Grant 1-005733. We would like to thank the Functional Genomics Center Zurich, and in particular Lennart Opitz and Dr. Maria Domenica Moccia, for their support on the RNA-seq data analysis and Dr. Roberto Fiore from the System Neuroscience Lab at ETH Zurich for providing the rat tails for the collagen-1 extraction. We further acknowledge Dr. Evi Masschelein for

performing the Achilles tenotomies and the Laboratory of Nutrition and Metabolic Epigenetics for the access to their Tapestation. We also thank our own Lab Technicians Barbara Niederöst and Maja Bollhalder for their practical and emotional support. Finally, we thank Dr. Knut Husmann and Dr. Annamari Katariina Alitalo for their help and support with obtaining the license for animal experimentation and animal husbandry.

## Additional information

### Funding

| Funder | Grant reference number | Author |
|---|---|---|
| ETH Zürich Foundation | 1-005733 | Katrien De Bock |

The funders had no role in study design, data collection and interpretation, or the decision to submit the work for publication.

### Author contributions

Tino Stauber, Conceptualization, Data curation, Software, Formal analysis, Validation, Investigation, Visualization, Methodology, Writing – original draft, Writing – review and editing; Greta Moschini, Investigation, Methodology, Writing – review and editing; Amro A Hussien, Conceptualization, Methodology, Writing – review and editing; Patrick Klaus Jaeger, Software, Formal analysis, Visualization, Methodology, Writing – review and editing; Katrien De Bock, Conceptualization, Resources, Supervision, Funding acquisition, Writing – review and editing; Jess G Snedeker, Conceptualization, Resources, Supervision, Funding acquisition, Project administration, Writing – review and editing

### Author ORCIDs

Tino Stauber ● https://orcid.org/0000-0002-4060-5747
Amro A Hussien ● https://orcid.org/0000-0002-9324-9360
Jess G Snedeker ● https://orcid.org/0000-0002-8115-0275

### Ethics

Tendon tissues from tendinopathic and normal control tendons were collected with informed consent including consent to publish from human patients undergoing treatment at the University Hospital Balgrist (ethical permission numbers 2016-02665 and 2020-0119 as approved by the institutional review board of the Canton of Zurich).

All experiments were approved by the responsible authorities (Canton Zurich license number ZH104-18 & ZH058-21).

Reviewer #1 (Public Review): https://doi.org/10.7554/eLife.87092.3.sa1
Author response https://doi.org/10.7554/eLife.87092.3.sa2

## Additional files

### Supplementary files

Supplementary file 1. Human patient microarray metadata. GEO accession number, patient sex, source tissue, patient age, donor number, and disease state of the isolated tissue ordered by GEO accession number. Samples from sheathed tendons are strikethrough and were excluded from further analysis.

MDAR checklist

Source code 1. R code file used for the human microarray analysis.

Source code 2. ImageJ code file used for the analysis of histological sections from humans.

Source code 3. R code file used for the analysis of histological sections from humans.

Source code 4. ImageJ code file used for the analysis of histological sections from assembloids.

Source code 5. R code file used for the analysis of histological sections from assembloids.

## Data availability

Sequencing data have been deposited in GEO under accession code GSE214015. The image and sequencing data analysis code (R and ImageJ) is included in the supporting files (Source code 1–5). Metadata on the human patients is included in the supporting files (*Figure 2—figure supplement 1*).

The following dataset was generated:

| Author(s) | Year | Dataset title | Dataset URL | Database and Identifier |
|---|---|---|---|---|
| Stauber T, Moschini G, Hussien AA, Jaeger PK, de Bock K, Snedeker JG | 2023 | Il-6 signaling exacerbates hallmarks of tendon lesions by stimulating progenitor proliferation & migration to damage | https://www.ncbi.nlm.nih.gov/geo/query/acc.cgi?acc=GSE214015 | NCBI Gene Expression Omnibus, GSE214015 |

The following previously published dataset was used:

| Author(s) | Year | Dataset title | Dataset URL | Database and Identifier |
|---|---|---|---|---|
| Jelinsky SA | 2010 | Analysis of Human Tendinopathy Gene Expression | https://www.ncbi.nlm.nih.gov/geo/query/acc.cgi?acc=GSE26051 | NCBI Gene Expression Omnibus, GSE26051 |

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
