## [Editor Report · eLife assessment]

This **important** study defines signaling mechanisms in tendinopathy development, which is significant as there is a clear need to identify therapeutic targets to prevent or reverse tendon pathology. The evidence supporting the conclusions are **compelling** combining an existing human tendinopathy transcriptomics dataset with ex-vivo assembloid model, and an in vivo injury model using genetic reporter mice. This work will be of interest to developmental and stem cell biologists.

---

## [Referee Report · Reviewer #1 (Public Review)]

This work by Stauber et al., is focused on understanding the signaling mechanisms that are associated with tendinopathy development, and by screening a panel of human tendinopathy samples, identified IL-6/JAK/STAT as a potential mediator of this pathology. Using an innovate explant model they delineated the requirement for IL-6 in the main body of the tendon to alter the dynamics of extrinsic fibroblasts. These studies are complemented by in vivo studies that include a Scx-GFP reporter. This approach facilitates examination of the effects of IL6-/- on Scx+ cells, and the differences observed between ex vivo and in vivo contexts.

The use of a publicly available existing dataset is considered a strength, since this dataset includes expression data from several different human tendons experiencing tendinopathy. The revised analysis that includes only non-sheathed tendons facilitates the identification of potentially conserved regulators of the tendinopathy phenotype, with immunostaining for CD90, IL-6R, and IL-6 expression in human tendinopathy samples providing important validation of the transcriptomic studies.

---

## [Author Response]

Author response:

The following is the authors’ response to the original reviews.

**eLife assessment:**
This important study details an enrichment of the IL-6 signaling pathway in human tendinopathy and applies transcriptional profiling to an advanced in vitro model to test IL-6 specific phenotypes in tendinopathy. Overall, the strength of evidence is solid yet incomplete, as transcriptomic measurements provide clarity, though functional studies including analysis of proliferation are needed to confirm these findings. This work will be of interest to stem cell biologists and immunologists.

To functionally assess the effect of IL-6 on Scx+ fibroblast proliferation in an acute injury, we repeated the in vivo studies with an EdU staining and a newly established IL-6 KO x ScxGFP+ mouse line. We found no evidence for this effect in acute injuries and acknowledge this in the revised manuscript.

We further added data collected by combining fluorescence microscopy with human patient-derived tissue to strengthen the link between IL-6, IL-6R, and proliferation of CD90+ cells in chronic injuries.

See comment 1.1.

See comment 2.4.

Changes:

- Title

- Abstract

- Figure 2 and 3 (new data)

- Figure 7 (new data)

- Results

- Discussion

**Reviewer 1**
(1.1) First, the experimental approach does not directly assess proliferation, as such the conclusions regarding proliferation are not well supported. In the ex-vivo model, the use of cell counting approaches is somewhat acceptable since the system is constrained by the absence of potential influx of new cells. However, given the nearly unlimited supply of extrinsically derived cells in vivo (vs. the explant model), assessment of actual proliferation (e.g. Edu, BrdU, Ki67) is critical to support this conclusion.

To assess the effect of IL-6 on Scx+ fibroblast proliferation in an acute injury, we repeated the in vivo studies with an EdU staining and a newly established IL-6 KO x ScxGFP+ mouse line to combat the considerable background noise of currently available Scx antibodies.

Under the improved design of these experiments, we could detect no effect of IL-6 on ScxGFP+ cells in an acute injury in vivo. We have therefore replaced figure 5 with the new results in figure 7 and moved figure 5F to the supplementary materials (Supplementary figure 9).

We acknowledge and discuss this in the discussion section.

See comment 2.4.

See comment 2.11.

Changes:

- Title

- Abstract

- Figure 7 (new data)

- Supplementary Figure 9

- Results

- Discussion

(1.2) Second, the justification for the use of Scx-GFP+ cells as a progenitor population is not well supported. Indeed, in the discussion, Scx+ cells are treated as though they are uniformly a progenitor population, when the diversity of this population has been established by the cited studies, which do not suggest that these are progenitor populations. Additional definition/ delineation of these cells to identify the subset of these cells that may actually display other putative progenitor markers would support the conclusions. As it stands, the study currently provides important information on the impact of IL6 on Scx+ cells, but not tendon progenitors.

We further delineated the extrinsic cell populations isolated from mouse Achilles tendons of ScxGFP+ mice using flow cytometric analysis and RT-qPCR. We used tendon population markers suggested by sc-RNA-seq of mouse Achilles tendons.

(De Micheli et al., Am. J. Physiol. - Cell Physiol., 2020, 319(5), DOI: 10.1152/ajpcell.00372.2020)

While a small subpopulation of these cells expressed typical progenitor markers (i.e. CD45 and CD146), we could detect no overlap with Scx+ cells. As suggested by the reviewer, we therefore replaced occurrences of “progenitor” in the manuscript with “fibroblast” and performed additional experiments with human patient-derived tissue sections and the fibroblast marker CD90.

See comment 2.1.

Changes:

- Title

- Abstract

- Figure 2 (new data)

- Figure 3 (new data)

- Supplementary Figure 6 (new data)

- Results

- Discussion

(1.3) Clarity regarding the relevance of the 'sheath-like' component of the assembloid would provide helpful context regarding which types of tendons are likely to have this type of communication vs. those that do not, and if there are differences in tendinopathy prevalence. Understanding why/how this communication between structures is relevant is important.

Our assembloid concept is inspired by the structure of unsheathed tendons (i.e. biceps, semitendinosus, gracilis) and not sheathed tendons like the flexor tendons.

We agree that clarity regarding the tendon type having this type of communication is important, so we sharpened previously blurry text passages in the revised manuscript.

Text changes:

- Introduction, page 3

- Results, page 4

- Results, page 8

- Results, page 9

- Results, page 11

- Discussion, page 25

- Discussion, page 26

- Experimental section, page 28

- Figure 1

- Figure 2

- Figure 3

- Supplementary Table 1

- Supplementary Figure 3

- Supplementary Figure 4

(1.4) Minor: in the text for Figure 6 (2nd paragraph), the comma in 19,694 is superscripted.

Corrections were made throughout the manuscript.

Text changes:

- Results, page 4

- Results, page 12

- Results, page 19

- Results, page 21

(1.5) Minor: The inclusion of the Scx-GFP mouse should be included in the schematic Figure 5.

The results presented in the previous draft did not feature tissues from ScxGFP mice but used a Scx-antibody to visually detect Scx+ cells. In anticipation of the revision process, we bred a new IL-6 KO x ScxGFP+ mouse line and repeated the experiment. As suggested by the reviewer, the new schematic figure 7 as well as the former figure 5 moved to the supplementary material now includes this mouse.

Figure changes:

- Supplementary Figure 9 (former figure 5)

- Figure 7

**Reviewer 2**
(2.1) One question that comes to mind is whether the fibroblast progenitors in the extrinsic sheath of Achilles tendon is similar to those surrounding the tail tendon. The similarity of progenitors between different tendons is assumed with this model. I would consider this to be a minor issue.

Tail tendon fascicles are thought to have a low number of reparative fibroblasts / progenitor cells because they lack a developed extrinsic compartment. Achilles tendons are supposed to have a higher number of reparative fibroblasts / progenitor cells, as their fascicles are surrounded by an extrinsic compartment.

To verify this here, we added a better characterization and comparison of the cell populations isolated from the tail tendon fascicles and the Achilles tendons.

First, we added representative light microscopy images of these cells at different timepoints after being cultured on tissue-culture plastic.

Second, we performed flow cytometric analysis not only on the freshly digested tail tendon fascicles and Achilles tendons, but also on the cultured cells at the timepoint when they would have been embedded into the assembloids.

Third, we compared the expression of population-specific markers in cells derived from tail tendon fascicle and Achilles tendons.

As expected, tail tendon fascicle-derived cell populations appeared to be more elongated than Achilles tendon-derived populations shortly after isolation. Similarly, the “maintenance” fibroblasts in healthy tendons are more elongated than the reparative fibroblasts in diseased ones. After culture and priming in tendinopathic niche conditions, both populations assumed a more roundish, reparative phenotype.

This was consistent with the flow cytometric analysis, which revealed a large difference between freshly isolated populations, that disappeared after extended culture and priming in tendinopathic niche conditions. Gene expression in tail tendon fascicle-derived and Achilles tendon-derived cells was similar after extended culture and priming in tendinopathic niche conditions.

See comment 1.2.

See comment 2.10.

Changes:

- Supplementary Figure 6 (new data)

- Results, page 11

(2.2) The authors use core tendons from IL-6 knockout mice and progenitors from wild-type mice. The reasoning behind this approach was a little confusing... is IL-6 expressed solely in the tendon core compared to the extrinsic sheath?

Insights gained from human patient-derived tissues (Figure 2) suggest that in a healthy tendon, most of the IL-6 is located in the extrinsic compartment but distributed over compartments in the tendinopathic ones.

Our assembloid design mimicks this by embedding wildtype fibroblasts into the extrinsic compartment. Our hypothesis was that a wildtype core in tendinopathic niche conditions attracts reparative fibroblasts through IL-6, while an IL-6 knock-out core does not. Therefore, it was important to establish IL-6 gradients close to what they seem to be in vivo*.*

Nevertheless, we have to acknowledge that the amount of IL-6 secreted by extrinsic fibroblasts in isolation is quite small compared to what is secreted by a wildtype core (Supplementary Figure 7). Attributing IL-6 in the supernatant of a WT core // WT fibroblast assembloid to the correct cell population is challenging but could be part of future research.

Changes:

- Figure 2 (new data)

- Supplementary Figure 7 (new data)

- Results, page 12

(2.3) Is a co-culture system for 7 days appropriate to model tendinopathy without the supplementation of exogenous inflammatory compounds? The transcriptomic differences in Figure 3 seem to be subtle, and may perhaps suggest that it could be a model that more closely resembles steady state compared to tendinopathy. If so, is IL-6 still relevant during steady state?

The collective experience in our lab is that core explants exposed to tendinopathic niche conditions (i.e. serum, 37°C, high oxygen, and high glucose levels) assume a disease-like phenotype. (i.e. Wunderli et al., Matrix Biology, 2020, Volume 89 https://doi.org/10.1016/j.matbio.2019.12.003 and Blache et al., Sci. Rep., 2021, 11(1), DOI 10.1038/s41598-021-85331-1).

Specifically for our core // fibroblast co-culture system, we have reported the emergence of exaggerated tendinopathic hallmarks in a previous publication (Stauber et al., Adv. Healthc. Mater., 2021, 10(20), https://doi.org/10.1002/adhm.202100741).

We clarified the use of previously validated tendinopathic niche conditions in this manuscript.

Changes:

- Introduction, page 3

- Results, page 12

(2.4) The results presented in Figures 4 and 5 are impressive, demonstrating a link between IL-6 and fibroblast progenitor numbers and migration. Their experimental design in these figures show strong evidence, using Tocilizumab and recombinant IL-6 to rescue shown phenotypes. I would reduce the claims on proliferation, however, unless a proliferation-specific marker (e.g., Ki67, BrdU, EdU) is included in confocal analyses of Scx+ progenitors.

As reviewer 1 pointed out as well, it is important to use a proliferation-specific marker “given the nearly unlimited supply of extrinsically derived cells in vivo (vs. the explant model)”.

To assess the effect of IL-6 on Scx+ fibroblast proliferation in vivo, we repeated those experiments with a proliferation-specific EdU staining and a newly established IL-6 KO x ScxGFP+ mouse line.

Under this improved design, we could not detect an effect of IL-6 on proliferation in an acute injury in vivo.

We have therefore replaced figure 5 with the new results in figure 7 and moved figure 5F to the supplementary materials (Supplementary figure 9).

We acknowledge and discuss this in the discussion section and softened our statements in the title and the abstract.

See comment 1.1.

See comment 2.11.

Changes:

- Title

- Abstract

- Figure 7 (new data)

- Supplementary Figure 9

- Results

- Discussion

(2.5) I think it would significantly strengthen the study if they could measure tendon healing in IL-6 knockouts or in wild-type mice treated with IL-6 inhibitors, since conventional ablation of IL-6 may lead to the elevation of compensatory IL-6 superfamily ligands that could activate STAT signaling. The authors claim that reducing IL-6 signaling decreases transcriptomic signatures of tendinopathy, but IL-6 may be necessary to promote normal healing of the tendon following injury. It is supposed that a lack of Scx+ progenitor migration would delay tendon healing.

Indeed, another study using the same IL-6 knock-out strain showed that a lack of IL-6 signaling resulted in slightly inferior mechanical properties in healing patellar tendons (Lin et al., J. Biomech., 39(1), 2006 https://doi.org/10.1016/j.jbiomech.2004.11.009)

Also, it might be due to the elevation of compensatory IL-6 superfamily ligands that we found no effect of IL-6 on the proliferation of Scx+ cells in an acute injury in vivo.

Therefore, assessing the effects of IL-6 inhibitors on tendon healing following an acute injury would have been of great interest to us. Unfortunately, getting the necessary permission from the animal experimentation office for a new invasive treatment protocol was outside of our scope due to the severity degree and time limitations.

We incorporated and acknowledged these important points in the discussion.

Text changes:

- Introduction, page 3

- Discussion, page 26

(2.6) Do IL-6 knockout mice and/or mice treated with IL-6 inhibitors have delayed healing following Achilles tendon resection? Please provide experimental evidence.

See comment 2.5.

(2.7) I would suggest reducing claims on proliferation, or include a proliferation specific marker (e.g., Ki67, BrdU, EdU) in confocal analyses of Scx+ progenitors.

See comment 1.1.

See comment 2.4.

(2.8) Supplementary Figures 1 and 2: the authors removed outliers. Please specify exactly which outliers were removed in the figures, and provide additional information on the criteria used to identify these outliers.

To address this comment, we sharpened our criteria for identifying outliers and re-did the analysis depicted in figure 1.

Briefly, we excluded 5 normal and 5 tendinopathic samples from sheathed tendons which have a different compartmental structure than unsheathed tendons.

A complete separate analysis of the sheathed tendons would have been beyond the scope of this manuscript, but early screening suggested that IL-6 transcripts are not increased in sheathed tendinopathic tendons.

We made text changes throughout the manuscript and to the supplementary table 1 and supplementary figure 2 to clearly state our criteria for excluding samples / outliers.

Changes:

- Introduction, page 3

- Results, page 4

- Results, page 8

- Results, page 9

- Results, page 11

- Discussion, page 25

- Discussion, page 26

- Experimental section, page 28

- Figure 1,

- Figure 2,

- Figure 3,

- Supplementary table 1,

- Supplementary figure 2,

- Supplementary figure 3,

- Supplementary figure 4,

(2.9) Whenever "positive enrichment" is mentioned in the text, please specify in what group. It is presumed that the enrichment, for example, in the first figure is associated with tendinopathy samples compared to controls, though it is a bit unclear.

The direction of the enrichment was added to the text.

Text changes:

- Abstract, page 1

- Introduction, page 3

- Results, page 4

- Results, page 6

- Results, page 12

- Results, page 14

- Results, page 19

- Results, page 21

- Discussion, page 25

- Discussion, page 26

- Discussion, page 27

- Figure 1

- Figure 5

- Figure 8

- Figure 9

- Supplementary figure 3

- Supplementary figure 4

- Supplementary figure 6

- Supplementary figure 8

- Supplementary figure 11

- Supplementary figure 12

- Supplementary figure 14

(2.10) Are tail tendon progenitors similar to Achilles tendon progenitors? Please provide a statement that shows similarity (in function, transcriptome, etc.) to support the in vitro tendon model.

See comment 1.2.

See comment 2.1.

(2.11) Are the results in Figure 5F significant? It seems that your pictures show a dramatic change in migration, but the quantification does not?

We repeated the in vivo studies with a newly established IL-6 KO x ScxGFP+ mouse line to combat the considerable background noise of currently available Scx antibodies.

Under the improved design of these experiments, we could not detect an effect of IL-6 on ScxGFP+ cells migration in an acute injury in vivo.

We have therefore replaced figure 5 with the new results in figure 7 and moved figure 5F to the supplementary materials (Supplementary figure 9)

We acknowledge and discuss this in the discussion section.

See comment 1.1.

See comment 2.4.

Changes:

- Title

- Abstract

- Figure 7 (new data)

- Supplementary Figure 9

- Results

- Discussion

(2.12) Please provide additional discussion points on cis- versus trans-IL6 signaling in your results found in mouse. Do you think researchers/clinicians would want to target trans-IL6 signaling based on your results? Please support these statements with the expression of IL6R on cells found in the tendon core and external sheath progenitors.

To address this comment, we performed flow cytometric analysis on Achilles tendon-derived fibroblasts expanded in 2D and digested sub-compartments of the assembloids (Supplementary Figure 7).

These data suggest that IL6R is neither expressed by core nor extrinsic fibroblasts, but mainly comes from core-resident CD45+ tenophages.

Human samples co-stained for IL6R and CD68 (an established human macrophage marker) confirmed macrophages as a source of IL-6R in vivo. However, human samples co-stained for IL6R and CD90 (an established marker of reparative fibroblasts in humans) also detected IL6R on CD90+ cells, which have not yet been reported to express IL6R themselves.

Overall, it is likely that trans-IL-6 signaling is more important for the activation of reparative fibroblasts than cis-IL-6 signaling. We added these statements to the manuscript.

Changes:

- Results, page 9

- Results, page 12

- Discussion, page 25

- Discussion, page 26

- Figure 3 (new data)

- Supplementary figure 7 (new data)

(2.13) Please provide more detail on collagen isolation from rat tail in the methods section.

We provided more details on collagen isolation from rat tail in the experimental section (page 29)

Changes:

- Experimental section, page 29

(2.14) Please comment on whether your in vitro system resembles tendinopathy or a steady state tendon. If it models more of a steady state system, would IL-6 still be relevant?

See comment 2.3.

Detailed feedback:
**Reviewer 1:**
This work by Stauber et al. is focused on understanding the signaling mechanisms that are associated with tendinopathy development, and by screening a panel of human tendinopathy samples, identified IL-6/JAK/STAT as a potential mediator of this pathology. Using an innovative explant model they delineated the requirement for IL-6 in the main body of the tendon to alter the dynamics of cells in the peritendinous synovial sheath space.The use of a publicly available existing dataset is considered a strength since this dataset includes expression data from several different human tendons experiencing tendinopathy. This facilitates the identification of potentially conserved regulators of the tendinopathy phenotype.The clear transcriptional shifts between WT and IL6-/- cores demonstrates the utility of the assembloid model, and supports the importance of IL6 in potentiating the cell response to this stimuli.
**Reviewer 2:**
The authors of this study describe a goal of elucidating the signaling pathways that are upregulated in tendinopathy in order to target these pathways for effective treatments. Their goal is honorable, as tendinopathy is a common debilitating condition with limited treatments. The authors find that IL-6 signaling is upregulated in human tendinopathy samples with transcriptomic and GSEA analyses. The evidence of their initial findings are strong, providing a clinically-relevant phenotype that can be further studied using animal models.Along these lines, the authors continue with an advanced in vitro system using the mouse tail tendon as the core with progenitors isolated from the Achilles tendon as the external sheath embedded in a hydrogel matrix. One question that comes to mind is whether the fibroblast progenitors in the extrinsic sheath of Achilles tendon is similar to those surrounding the tail tendon. The similarity of progenitors between different tendons is assumed with this model. I would consider this to be a minor issue, and would consider the in vitro system to be an additional strength of this study.In order to address the IL-6 signaling pathway, the authors use core tendons from IL-6 knockout mice and progenitors from wild-type mice. The reasoning behind this approach was a little confusing... is IL-6 expressed solely in the tendon core compared to the extrinsic sheath? Furthermore, is a co-culture system for 7 days appropriate to model tendinopathy without the supplementation of exogenous inflammatory compounds? The transcriptomic differences in Figure 3 seem to be subtle, and may perhaps suggest that it could be a model that more closely resembles steady state compared to tendinopathy. If so, is IL-6 still relevant during steady state?Nevertheless, the results presented in Figures 4 and 5 are impressive, demonstrating a link between IL-6 and fibroblast progenitor numbers and migration. Their experimental design in these figures show strong evidence, using Tocilizumab and recombinant IL-6 to rescue shown phenotypes. I would reduce the claims on proliferation, however, unless a proliferation-specific marker (e.g., Ki67, BrdU, EdU) is included in confocal analyses of Scx+ progenitors. The Achilles tendon injury model provides a nice in vivo confirmation of Scx-progenitor migration to the neotendon.Given their goal to elucidate signaling pathways that could be targeted in the clinic, I think it would significantly strengthen the study if they could measure tendon healing in IL-6 knockouts or in wild-type mice treated with IL-6 inhibitors, since conventional ablation of IL-6 may lead to the elevation of compensatory IL-6 superfamily ligands that could activate STAT signaling. The authors claim that reducing IL-6 signaling decreases transcriptomic signatures of tendinopathy, but IL-6 may be necessary to promote normal healing of the tendon following injury. It is supposed that a lack of Scx+ progenitor migration would delay tendon healing.Overall, the authors of this study elucidated IL-6 signaling in tendinopathy and provided a strong level of evidence to support their conclusions at the transcriptomic level. However, functional studies are needed to confirm these phenotypes and fully support their aims and conclusions. With these additional studies, this work has the potential to significantly influence treatments for those suffering from tendinopathy.